# Prolonged experimental CD4+ T-cell depletion does not cause disease progression in SIV-infected African green monkeys

Quentin Le Hingrat [1,2], Paola Sette[1,2], Cuiling Xu[1,2], Andrew R. Rahmberg [3], Lilas Tarnus[1,2], Haritha Annapureddy[1], Adam Kleinman[2], Egidio Brocca-Cofano[1], Ranjit Sivanandham [1], Sindhuja Sivanandham [1], Tianyu He[1], Daniel J. Capreri[1], Dongzhu Ma[1,2], Jacob D. Estes[4,5], Jason M. Brenchley[3], Cristian Apetrei [2,6] & Ivona Pandrea [1,6] ✉

CD4+ T-cell depletion is a hallmark of HIV infection, leading to impairment of cellular immunity and opportunistic infections, but its contribution to SIV/HIV-associated gut dysfunction is unknown. Chronically SIV-infected African Green Monkeys (AGMs) partially recover mucosal CD4+ T-cells, maintain gut integrity and do not progress to AIDS. Here we assess the impact of prolonged, antibody-mediated CD4 + T-cell depletion on gut integrity and natural history of SIV infection in AGMs. All circulating CD4+ T-cells and >90% of mucosal CD4+ T-cells are depleted. Plasma viral loads and cell-associated viral RNA in tissues are lower in CD4+-cell-depleted animals. CD4+-cell-depleted AGMs maintain gut integrity, control immune activation and do not progress to AIDS. We thus conclude that CD4+ T-cell depletion is not a determinant of SIV-related gut dysfunction, when gastrointestinal tract epithelial damage and inflammation are absent, suggesting that disease progression and resistance to AIDS are independent of CD4+ T-cell restoration in SIVagm-infected AGMs.

Progressive HIV/SIV infections are characterized by gastrointestinal (GI) tract dysfunction and massive depletion of CD4+ T-cells from circulation and tissues [i.e., GI mucosa and lymph nodes (LNs)] as early as 7 days postinfection (dpi)[1,2]. Loss of gut lamina propria CD4+ T-cells coincides with the peak of viral replication at mucosal sites and immediately precedes the detection of microbial products in circulation, due to the loss of GI tract integrity[3]. Multiple virally-induced mechanisms lead to the depletion of CD4+ T-cells[4], including death of infected cells through the action of HIV/SIV-specific CD8+ T-cells[5], cytolysis induced by virus release[6] and programmed cell death[7,8]. However, most of CD4+ T-cells are not HIV/SIV-infected[9] and other mechanisms are involved: (i) increased apoptosis of uninfected cells exposed to viral antigens[10,11], (ii) activation-induced cell death[12], (iii) pyroptosis in cells undergoing abortive infection[13], and (iv) destruction of cells trapped in neutrophil extracellular traps induced by HIV/SIV infection[14,15]. CD4+ T-cell depletion is thus central to HIV/SIV infection, and, despite the unfolding of antiretroviral therapy (ART), CD4+ T-cells are not entirely restored in people living with HIV (PWH) receiving ART[16]. Most notably, CD4+ T-cell restoration is only partial in the GI tract, even in early-treated PWH[17].

Multiple studies investigated the impact of various parameters on the progression of untreated HIV/SIV infection, based on the rationale

[1]Department of Pathology, School of Medicine, University of Pittsburgh, Pittsburgh, PA, USA. [2]Division of Infectious Diseases, Department of Medicine, School of Medicine, University of Pittsburgh, Pittsburgh, PA, USA. [3]Barrier Immunity Section, Lab of Viral Diseases, Division of Intramural Research, NIAID, NIH, Bethesda, MD, USA. [4]Vaccine and Gene Therapy Institute, Oregon Health & Science University, Portland, OR, USA. [5]Division of Pathobiology and Immunology, Oregon National Primate Research Center, Oregon Health & Science University, Portland, OR, USA. [6]Department of Infectious Diseases and Microbiology, Graduate School of Public Health, University of Pittsburgh, Pittsburgh, PA, USA. ✉e-mail: pandrea@pitt.edu

that identifying key driver(s) of disease progression could help establish treatment priorities. The role of systemic inflammation and T-cell activation in fueling viral replication and CD4[+] T-cell depletion is widely acknowledged[18], but the exact impact of CD4[+] T-cell depletion on gut damage and disease progression remains to be determined. Peripheral CD4[+] T-cell counts predict disease progression in untreated PWH[19,20], and decreasing CD4[+] T-cell counts are associated with the onset of opportunistic infections and AIDS[21].

CD4[+] T-cell depletion itself can impact disease progression. Preservation of the Th17 subset of CD4[+] T-cells could help maintain GI tract integrity and limit local and systemic inflammation[22,23]. CD4[+] T-cell depletion can also impair SIV-specific cytotoxic and humoral responses, leading to loss of virus control[24]. Furthermore, CD4[+] T-cell destruction fuels local inflammation by releasing proinflammatory products during cell death[25].

Experimental CD4[+]-cell depletion in rhesus macaques (RMs) prior to SIV infection abrogated the postacute control of viremia, led to the emergence of CD4-independent envelopes and to a larger number of non-T-infected cells, and accelerated disease progression[26,27]. Meanwhile, administration of a single dose of this CD4-depleting antibody to chronically SIV-infected sooty mangabeys resulted in decreased plasma viral loads (pVLs) that correlated with the number of residual CD4[+] T-cells[28]. A slight reduction in the size of viral reservoir occurred in ART-treated, SIV-infected RMs after administrating 6 doses of this CD4-depleting antibody[29].

To directly assess the role of CD4[+] T-cells in SIV disease progression and establish whether their preservation in natural hosts of SIV contributes to preventing intestinal dysfunction and protecting from disease progression, we induced a prolonged (>1 year) CD4[+] T-cell depletion in AGMs. SIV infection is generally nonpathogenic in AGMs, despite a massive peripheral and mucosal CD4[+] T-cell depletion during acute infection[30–32]. During chronic infection, there is a partial recovery of circulating and mucosal CD4[+] T-cells[33], and we have shown that AGMs maintain GI tract integrity throughout the SIV infection, limiting microbial translocation and thus avoiding fueling inflammation and T-cell activation[33,34]. Here, we report that prolonged antibody-mediated depletion of CD4-expressing cells does not alter AGMs' ability to preserve their GI tract integrity and keep chronic inflammation and immune activation at bay.

As such, we show that, in the absence of epithelial gut damage, persistent inflammation and/or immune activation, CD4[+] T-cell depletion alone is not sufficient to induce gut dysfunction or disease progression. Furthermore, resistance to AIDS in natural hosts of SIV is independent of CD4[+] T-cells counts and/or their swift and robust restoration in blood and tissues.

## Results

### Study design
To assess the role of CD4[+]-cell depletion in driving gut dysfunction and HIV/SIV disease progression, we induced a systemic and prolonged antibody-mediated CD4[+]-cell depletion in AGMs. Twelve AGMs intravenously infected with 300 TCID$_{50}$ of SIVsab92018 (Fig. 1a) were divided into 2 groups. The first group ($n = 6$) received 21 doses of rhesusized depleting anti-CD4 monoclonal antibody (mAb) (CD4R1, clone OKT4A, NIH Nonhuman Primate Reagent Resource, Boston, MA), administered subcutaneously at 50 mg/kg at 21, 35 and 49 dpi, then every 3 weeks up to 427 dpi (21 doses in total). AGMs were euthanized around 454 dpi (i.e., after 433 days of experimental CD4[+]-cell depletion). The remaining AGMs ($n = 6$) served as controls, they did not receive CD4R1, and had similar sampling and testing schedules.

### Toxicity
Potential toxicity of CD4R1 was carefully monitored in the 6 AGMs that were infused with CD4R1, and no serious adverse events were observed. Furthermore, no hematological toxicity was detected

throughout the follow-up (Supplementary Fig. S1A–D). Similarly, neither renal, nor liver toxicity were evident from blood chemistry panels (Supplementary Fig. S1E–L). Hematoxylin and eosin (H&E) stains on multiple tissues obtained at necropsy did not reveal any opportunistic infection. A moderate microsteatosis was seen in the liver of CD4[+]-cell-depleted AGMs. However, liver enzymes remained in the normal range in all CD4R1-treated AGMs (Supplementary Fig. S1).

### CD4[+] T-cell dynamics
CD4[+] T-cell levels in different body compartments were measured flow-cytometrically (Fig. 1b). Prior to infection and before administration of CD4R1, circulating CD4[+] T-cell counts were similar between the CD4[+]-cell-depleted AGMs and controls ($241 \pm 70$ and $248 \pm 106$ cells/µL, respectively). In both groups, SIV-induced CD4[+] T-cell depletion reached a nadir at 10 dpi ($134 \pm 26$ and $116 \pm 48$ cells/µL, respectively), with an average depletion of 43% and 52%, respectively (Fig. 1c, d). In both groups, a slight recovery in peripheral CD4[+] T-cell counts occurred immediately before initiating the CD4R1 treatments. In controls, this steady restoration continued throughout the follow-up, plateauing at 75–80% of preinfection values (Fig. 1c, d). In CD4[+]-cell-depleted AGMs, a tenfold reduction in CD4[+] T-cell counts was observed after the first CD4R1 administration, from $197 \pm 55$ cells/µL at 21 dpi to $19 \pm 11$ cells/µL at 35 dpi (Fig. 1c). Subsequent infusions strengthened this substantial depletion, with a mean CD4[+] T-cell count of 3.4 cells/µL for samples taken after 48 dpi (interquartile range (IQR): 1.1–4.5 cells/µL). This translates into a depletion of 98.5% (IQR: 98.0–99.6%) of circulating CD4[+] T cells, compared to the preinfection levels (Fig. 1d).

This massive CD4[+] T-cell depletion was not limited to circulation. Significant differences were also observed between the two groups in the percentage of CD4[+] T-cells among lymphocytes isolated from superficial LNs and intestinal biopsies (duodenum and colon) (Fig. 1e–g). In the sLNs, SIV infection itself led to a decrease in the percentage of CD4[+] T-cells, with an average depletion of 19% during chronic SIV infection in controls (Fig. 1e). In sLNs of CD4[+]-cell-depleted AGMs, the repeated CD4R1 infusions progressively deepened the CD4[+] T-cell depletion, with a 5-fold reduction of CD4[+] T-cells at 84 dpi, and a 94% depletion after 150 dpi (Fig. 1h). Both naïve and memory subsets of CD4[+] T-cells were depleted in blood and sLNs of CD4R1-treated AGMs, with a preferential depletion of naïve T-cells in sLNs (Supplementary Fig. 2).

Duodenal and colonic biopsies were collected to evaluate the magnitude of the CD4[+] T-cell depletion in the small and large intestine, respectively, and to explore whether CD4[+] T-cell depletion uniformly occurred in these compartments. Prior to SIV infection, the percentage of lamina propria CD4[+] T-cells was higher in colon than in duodenum in both groups (7.7% versus 3.0% for CD4[+]-cell-depleted AGMs, and 7.7% versus 4.9% for controls) (Fig. 1f, g). A marked depletion of duodenal CD4[+] T-cells occurred between 10 and 21 dpi in both groups (Fig. 1f). At 10 dpi, mean duodenal CD4[+] T-cell depletion was 32% in controls and 50% in the study group ($p = 0.4$) (Fig. 1l). At 21 dpi, average CD4[+] T-cell depletion was 65% in controls and 47% in the study group ($p = 0.2$). In controls, nadir was reached around 73 dpi, with a 78% reduction from baseline (range: 66–85%). Then, a partial duodenal CD4[+] T-cell restoration occurred in controls, with a mean CD4[+] T-cell depletion of 57% after 150 dpi. At 73 dpi, duodenal CD4[+] T-cell depletion was not significantly different between CD4[+]-cell-depleted AGMs (average depletion: 84%, range: 77–95%) and controls ($p = 0.056$). However, there was no duodenal CD4[+] T-cell restoration in CD4R1-treated AGMs during chronic infection. On the contrary, depletion was consolidated, reaching 89% (range: 62–99%) after 150 dpi (Fig. 1l).

While colonic CD4[+] T-cell depletion persisted up to 305 dpi in controls, without any significant restoration (average CD4[+] T-cell depletion: 60.8%, range: 8.8–91.4%), depletion was even more prominent in CD4[+]-cell-depleted AGMs (average: 97.3%, range:

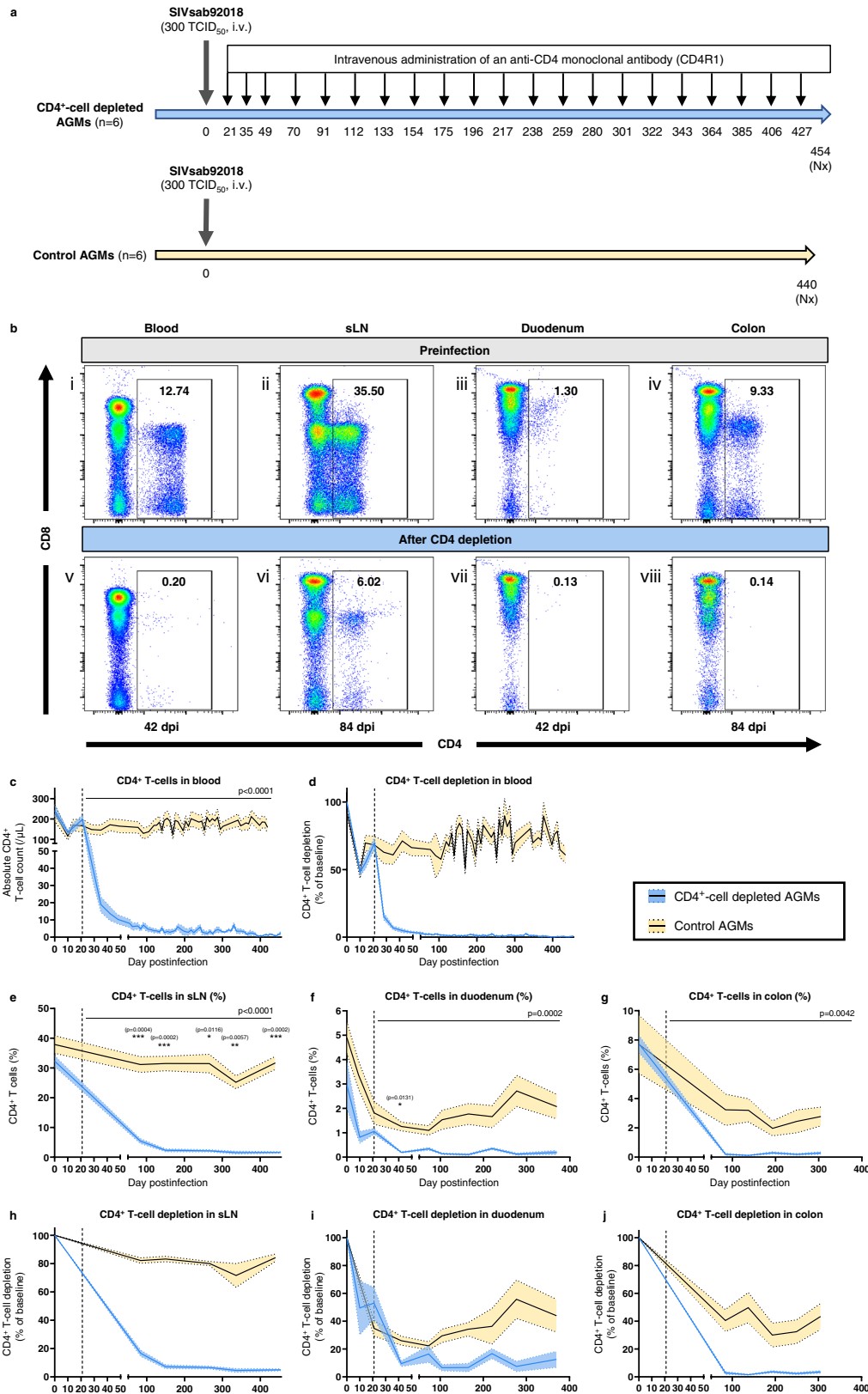

92.5–99.6%), and more profound in colon than in duodenum, with no CD4[+] T-cell restoration being observed throughout the follow-up (Fig. 1f–g, j).

Flow cytometry on tissues collected at necropsy also highlighted a massive CD4[+] T-cell depletion in jejunum, ileum, mesenteric LNs and liver of CD4[+]-cell-depleted AGMs, compared to controls (Fig. 2a). The magnitude of CD4[+] T-cell depletion in these tissues was in the range observed in superficial LNs and duodenum, i.e., ≈90% (Fig. 2b). Immunohistochemistry (IHC) for CD4[+] T-cells confirmed the efficacy of the CD4R1 antibody in depleting CD4[+] T-cells in the GI tract, with only residual CD4[+] T-cells left, notably in Peyer's patches and lymphoid aggregates (Fig. 2c–f).

**Fig. 1 | Dynamics of CD4⁺ T-cells in different tissues. a** Dynamics of CD4⁺ T-cells was assessed in SIVsab92018-infected AGMs with or without an experimental CD4⁺-cell depletion over a 430 days follow-up. A CD4-depleting antibody (CD4R1) was administered at 21, 35 and 49 dpi, then every 3 weeks, for a total of 21 infusions. **b** Representative images of flow cytometry of CD3⁺ T-cells in blood (i, v), sLNs (ii, vi) and intestinal biopsies (iii, iv, vii, viii) of one CD4⁺-cell-depleted AGM (AGM81), preinfection (i, ii, iii, iv) and postinfection and CD4⁺-cell depletion (v, vi, vii, viii). **c–j** Dynamics of CD4⁺ T-cells in CD4⁺-cell-depleted (blue) and control (yellow) AGMs in different tissues: blood (**c, d**), superficial LNs (**e, h**), duodenum (**f, i**) and colon (**g, j**). CD4⁺ T-cell counts are presented as absolute cell counts (**c**) or percentages of CD3⁺ T-cells (**e, f, g**). Magnitude of CD4⁺ T-cell depletion is represented with the percentage of CD4⁺ T-cell depletion from baseline (**d, h, i, j**). The dashed line indicates

the initiation of the CD4⁺-cell-depleting antibody in the study group. Data presented in (**c**)–(**j**) are means (solid lines) and standard error of the means (shaded regions within the dashed lines), with $n = 6$ animals per group. Differences between the two groups in the levels of CD4⁺ T-cells in blood and tissues after administering the CD4-depleting antibody were assessed using mixed-effects models (**c, e–g**), with Holm-Šídák's correction for multiple comparisons. For each tissue, mixed-effects model analyses revealed that CD4⁺-cell depletion had a significant effect on the levels of CD4⁺ T-cells. The corresponding p value is reported in the top right corner (**c, e–g**). Asterisks indicate timepoints for which differences were significant between the two groups, with *$p < 0.05$; **$p < 0.01$ and ***$p < 0.001$. AGMs African Green Monkeys, i.v. intravenous, Nx necropsy, sLN superficial lymph node, TCID₅₀ 50% tissue culture infectious dose.

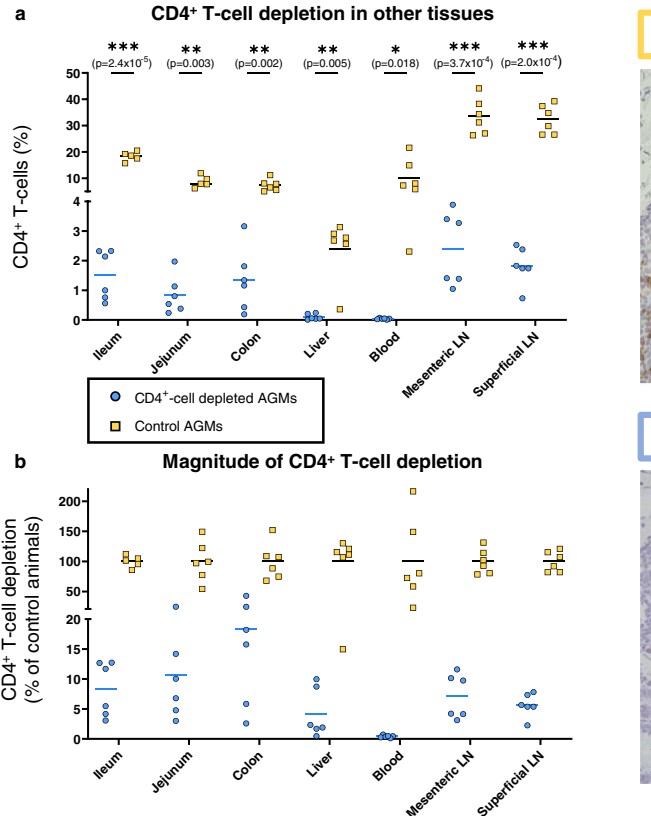

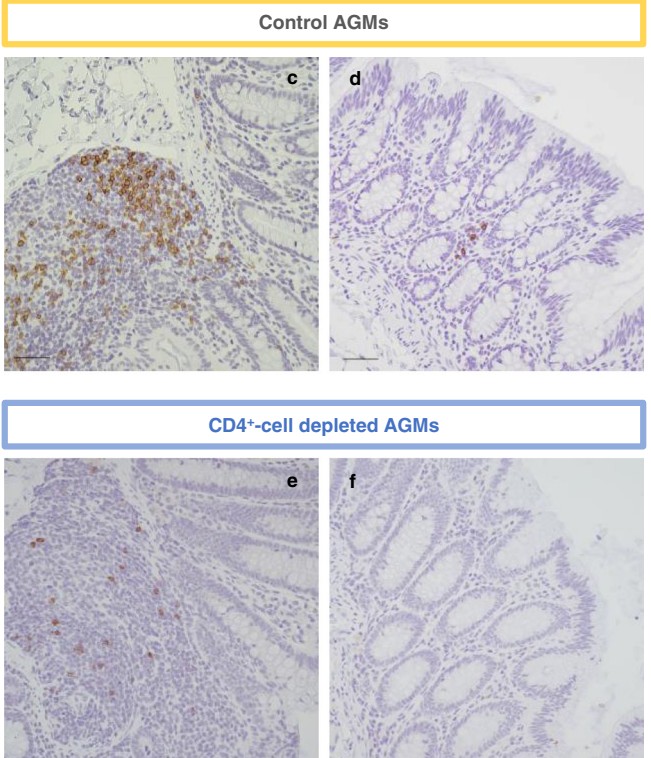

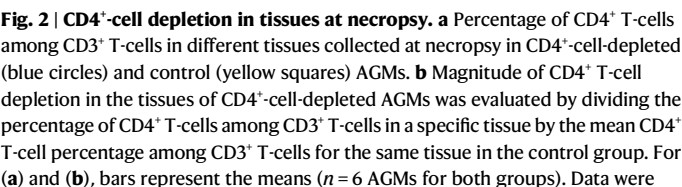

**Fig. 2 | CD4⁺-cell depletion in tissues at necropsy. a** Percentage of CD4⁺ T-cells among CD3⁺ T-cells in different tissues collected at necropsy in CD4⁺-cell-depleted (blue circles) and control (yellow squares) AGMs. **b** Magnitude of CD4⁺ T-cell depletion in the tissues of CD4⁺-cell-depleted AGMs was evaluated by dividing the percentage of CD4⁺ T-cells among CD3⁺ T-cells in a specific tissue by the mean CD4⁺ T-cell percentage among CD3⁺ T-cells for the same tissue in the control group. For (**a**) and (**b**), bars represent the means ($n = 6$ AGMs for both groups). Data were

analyzed with unpaired *t* tests with Welch correction using the Holm-Šídák method for multiple comparisons. Asterisks indicate statistical significance, with *$p < 0.05$; **$p < 0.01$ and ***$p < 0.001$. **c–f** Representative images showing IHC staining for CD4 expression (brown) in the GI tract of controls (**c, d**) and CD4⁺-cell-depleted (**e, f**) AGMs. Images of Peyer's patches (**c, e**) and lamina propria (**d, f**) are shown. On all IHC images, scale bar is 50 μm. These pictures are representative of the 6 CD4⁺-cell-depleted AGMs.

## Other immune cell subsets

In AGMs, immune functions of CD4⁺ T-cells can be supported, at least partially, by other immune cell subsets, notably CD8αα (or CD8α^low that arise via downregulation of CD4 by CD4⁺ T-cells) and double-negative (DN) T cells[35–37]. Thus, to assess whether functions of CD4⁺ T-cells were taken over by other immune cells, we investigated whether these subsets increased in number or in functionality in CD4⁺-cell-depleted AGMs. We also monitored the dynamics of monocytes, as this subset was increased in RMs treated with CD4R1 prior to SIVmac infection[27].

No increase in the absolute monocyte and CD8αα counts were observed in CD4⁺-cell-depleted AGMs (Fig. 3a, b). At baseline, absolute counts of DN T-cells were significantly higher in CD4⁺-cell-depleted AGMs than in controls (261 cells/μL vs 106 cells/μL; $p = 0.002$) but their

frequencies among CD3⁺ T-cells were similar between both groups (6.6% and 6.7%, respectively; $p = 0.94$) (Fig. 3c). DN T-cell counts were stable during the first 100 dpi, then slightly increased in both groups. Finally, when comparing the fractions of DN or CD8αα T-cells in sLNs and duodenal biopsies of CD4⁺-cell-depleted and control AGMs, no significant changes were observed (Fig. 3d–g).

## Functional profiling of T-cell subsets

To assess if the functionality of the different T-cell subsets changed when CD4⁺ T-cells were experimentally depleted, cells were stimulated with PMA and ionomycin to determine their expression of seven functional markers. The only significant change that occurred over time in CD4⁺-cell-depleted AGMs was an increase in the percentage of FoxP3-expressing CD4⁺ T-cells between preinfection and 21 dpi

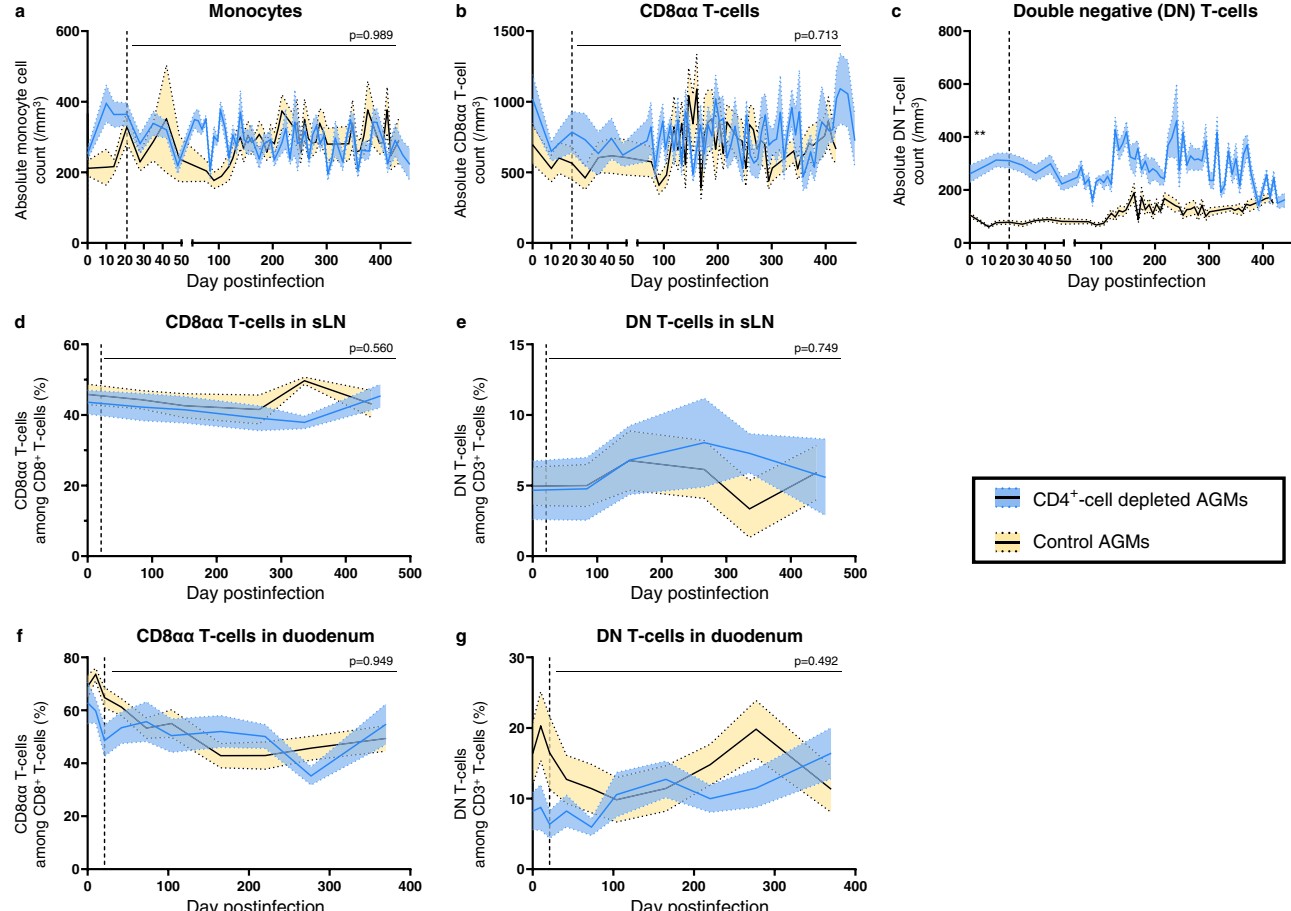

**Fig. 3 | Dynamics of other immune cell populations.** Dynamics of different immune cell subsets in circulation were monitored in CD4+-cell-depleted (blue) and control (yellow) AGMs, using flow cytometry. Monocytes (**a**), CD8αα (**b**) and double-negative (CD4negCD8neg) (**c**) T-cells were expressed as absolute cell counts per μL of peripheral blood. Dynamics of CD8αα (**d, f**) and double-negative (**e, g**) T-cells were also monitored in sLNs (**d–e**) and in duodenal biopsies (**f–g**). Values presented are means (solid lines) and standard error of the means (shaded regions within the dashed lines), with n = 6 AGMs in each group. When baseline levels were similar, differences between the two groups in the mean levels of different immune cell subsets after administering the CD4-depleting antibody were assessed using mixed-effects models, with Holm-Šídák's correction for multiple comparisons (**a, b** and **d–g**). Mixed-effects analysis revealed that CD4+-cell depletion did not have a significant effect on the level of those immune cell subsets and the corresponding p value is reported in top right corner of (**a, b**) and (**d–g**). For (**c**), absolute counts of DN T cells differed between the two groups at baseline (unpaired, two-sided, Mann–Whitney test). Asterisk indicates significant difference, with **p = 0.002. DN T-cells double-negative (CD4neg CD8neg) T-cells, sLNs superficial lymph nodes.

(p = 0.03). However, differences between T-cell subsets were apparent (Fig. 4a). When combining all timepoints, CD4+ T-cells significantly differed from CD8αα and CD8αβ T-cells. CD8αα differed from CD8αβ T cells regarding their expression of FoxP3, CD40L, and IL-2, but not regarding granzyme B, CD107a, IL-17 and IFN γ expression, reflecting the ability of CD8αα T-cells to assume characteristics of both CD4+ and CD8αβ T-cells. Controls exhibited broadly similar trends with few differences within T-cell subsets across timepoints and substantial differences between T-cell subsets (Fig. 4b). No significant differences were observed between controls and CD4+-cell-depleted AGMs regarding the fraction of cells expressing specific functional markers.

When analyzed for polyfunctionality, there were also no significant differences within each subset across timepoints, as determined by a partial permutation test performed in SPICE[38] (Supplementary Tables 1 and 2). However, in CD4+-cell-depleted AGMs, CD4+ T-cells did significantly differ from both CD8αα and CD8αβ T-cells at each timepoint. CD8αα and CD8αβ T-cells significantly differed at 0 and 371 dpi, but not at 21 or 42 dpi, again reflecting the intermediate nature of CD8αα T-cells (Supplementary Fig. 3). Controls displayed similar trends in polyfunctionality, with no significant differences across timepoints within a T-cell subset, but significant differences between the T-cell subsets (Supplementary Fig. 4).

## Slight impairment of the lymphocyte renewal in CD4+-cell-depleted AGMs

As CD4R1 induced a significant depletion of CD4+ T-cells in multiple tissues, including a preferential depletion of naïve CD4+ T-cells in sLNs, we assessed whether CD4+-cell depletion impacted the proliferative capacity of immune cells by quantifying the expression of the proliferation marker Ki-67.

A brief, 2-fold increase in CD4+ T-cell proliferation occurred in both groups at 10 dpi, before returning to baseline levels at 14 dpi in all AGMs. However, while the fraction of Ki-67-expressing circulating CD4+ T-cells did not vary after 14 dpi in controls, it surged again between 28 and 42 dpi in CD4+-cell-depleted AGMs, i.e., after the first CD4R1 infusion (Fig. 5a). CD4+ T-lymphocyte renewal could not be assessed for later timepoints, due to a limited number of circulating cells in CD4+-cell-depleted AGMs. In both groups, the proliferative capacity of CD4+ T-cells isolated from LNs slightly increased postinfection, but there were no significant differences between the two groups in the magnitude of CD4+ T-cell proliferation in sLNs (Fig. 5d).

Peripheral CD8+ T-cell proliferation also increased early during infection, but it peaked later, at 21 dpi (Fig. 5b). Then, a sharp decline in the fraction of proliferating circulating CD8+ T-cells occurred in both groups to nearly baseline levels by 73 dpi. The drop in the

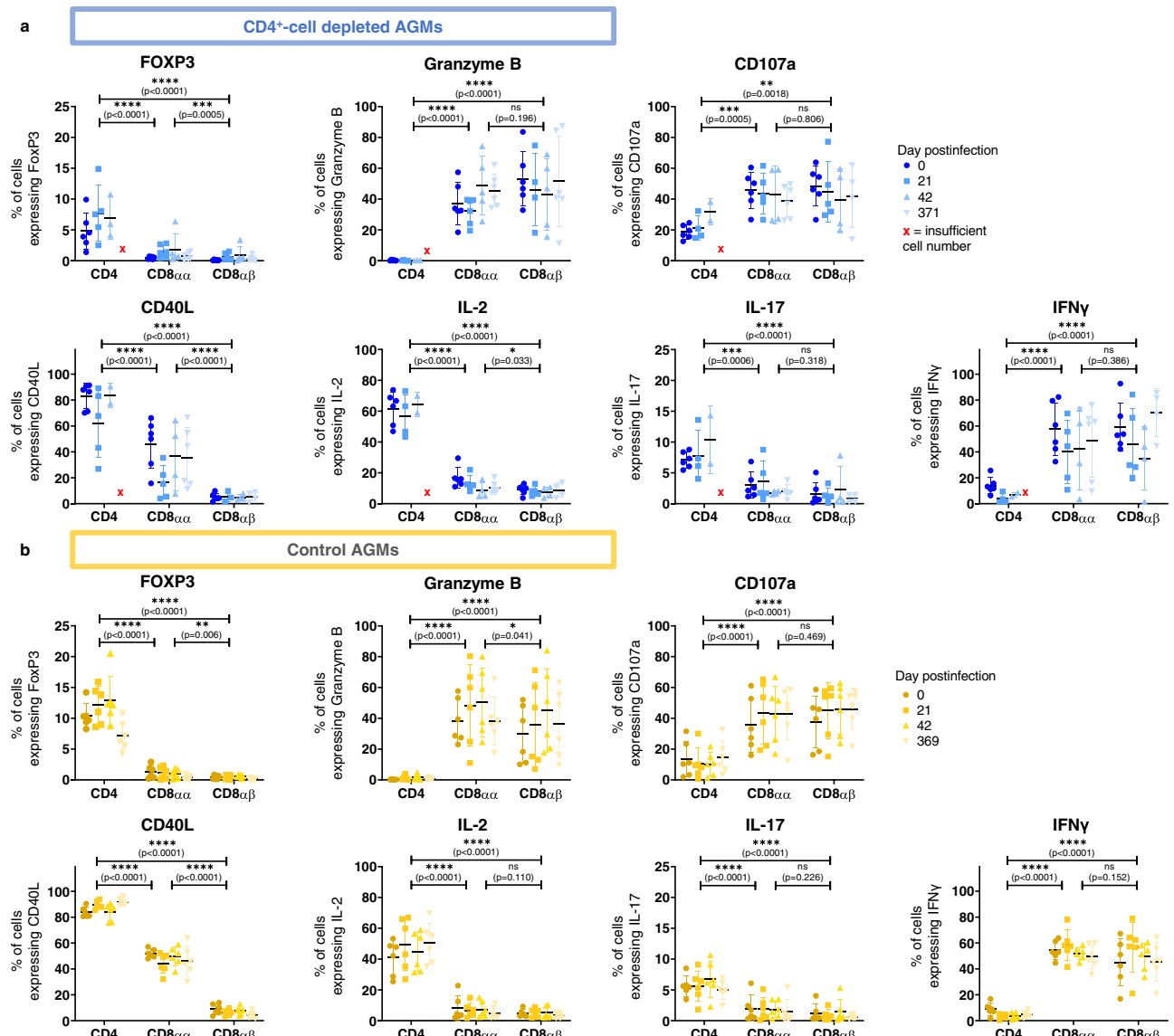

**Fig. 4 | Immune functions of T-cells.** Expression of 7 different functional markers at different timepoints pre and postinfection in different T-cell subsets in CD4⁺-cell-depleted AGMs (blue, **a**) and controls (yellow, **b**). Means ± standard deviations are displayed, with n = 6 AGMs for all points, except when less than 200 cells were acquired. Indicated statistics are comparisons between each T-cell subset when combining all timepoints. Data were analyzed with paired mixed-effects analysis followed with Tukey's multiple comparisons test. Asterisks indicate statistical significance. Abbreviations: *p < 0.05; **p < 0.01; ***p < 0.001 and ****p < 0.0001.

fraction of CD8⁺ T-cells expressing Ki-67 was slightly faster in CD4⁺-cell-depleted AGMs than in controls. IHC staining of sLNs also showed an increased in the fraction of Ki-67-expressing cells in the T-cell zone of sLNs, from 1.96% preinfection to 5.46% at 150 dpi in CD4⁺-cell-depleted AGMs, and from 2.63% at baseline to 4.02% at 150 dpi in controls (Fig. 5f).

The B cells expressing Ki-67 showed a stark increase during acute infection, from 8.2% preinfection to 17.0% at 21 dpi in CD4⁺-cell-depleted AGMs, and from 9.4% preinfection to 18.0% at 21 dpi in controls (Fig. 5c). B-cell proliferation peaked later, around 120 dpi in both groups (Fig. 5c). Like the CD8⁺ T-cells, the decrease in Ki-67-expressing CD20⁺ cells was steeper in CD4⁺-cell-depleted AGMs than in controls, yet the return to baseline levels was similar in both groups during chronic infection. IHC confirmed the increased Ki-67 expression in B follicles, from 1.57% preinfection to 16.98% at 150 dpi in CD4⁺-cell-depleted AGMs, and from 5.78% preinfection to 17.53% at 150 dpi in controls (Fig. 5h). A slight decrease was observed in necropsy tissues in both groups, although B-cell proliferation remained higher than at baseline (Fig. 5h–p).

Immune cell proliferation was also assessed in colon, both in lamina propria and the epithelial crypts (Fig. 6a). No significant increase in the overall Ki-67 expression by enterocytes in the intestinal crypts was observed between preinfection and necropsy in either group (Fig. 6a). Meanwhile, a moderate increase in the fraction of proliferating cells was observed at necropsy in the lamina propria of the controls, although it did not reach statistical significance (p value: 0.052) (Fig. 6b). We also assessed the fraction of proliferating CD8⁺ T-cells in colon and duodenal biopsies. No differences were observed between groups regarding CD8⁺ T-cell proliferation in colon biopsies, while Ki-67 expression by CD8⁺ T-cells increase in the duodenal biopsies collected at 21 and 42 dpi from controls, but not in those from CD4⁺-cell-depleted AGMs (Supplementary Fig. S5A, B).

**Experimental CD4⁺-cell depletion does not significantly impact the activation status of the CD8⁺ T-cells**

Persistent T-cell immune activation is a landmark of chronic pathogenic SIV and HIV infections[39]. Immune activation is reduced in

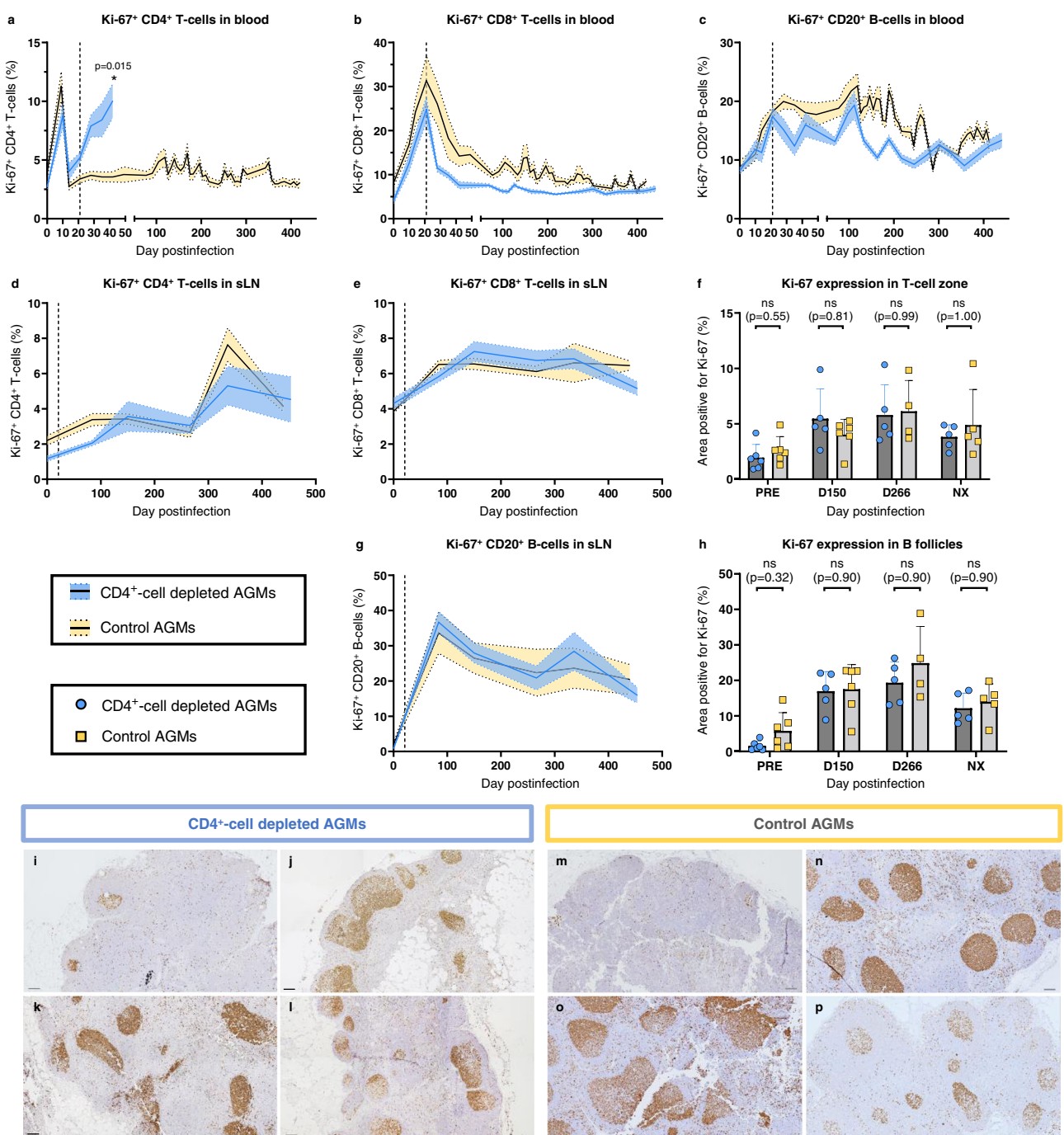

**Fig. 5 | T-cell and B-cell proliferation in circulation and superficial lymph nodes.** Dynamics of Ki-67 expression were monitored in CD4⁺ T-cells (**a**), CD8⁺ T-cells (**b**) and CD20⁺ B-cells (**c**) in the blood of CD4⁺-cell-depleted AGMs (blue) and controls (yellow). Ki-67 expression was determined in sLNs by flow cytometry (**d–e, g**) and by immunohistochemistry (IHC) (**f, h**). Ki-67 expression was monitored by flow cytometry in CD4⁺ T-cells (**d**), CD8⁺ T-cells (**e**) and CD20⁺ B-cells (**g**) isolated from sLN. Using IHC, cell proliferation was quantified in two different zones of the sLNs: the T-cell zones (**f**) and B follicles (**h**). Quantifications were performed using the FIJI software. Data presented are means and standard errors of the means (**a–e, g**) or means and standard deviations (**f, h**). For (**a–h**), 6 AGMs per group were analyzed, except for (**f**) and (**h**) for which 4–6 lymph nodes of different AGMs were analyzed in each group, depending on tissue availability. Data were analyzed with unpaired, two-sided, nonparametric Mann–Whitney tests, followed by Holm-Šídák's correction for multiple comparisons. Asterisk indicates statistical significance, with *$p < 0.05$. Representative IHC images of sLN of CD4⁺-cell-depleted (**i–l**) and control (**m–p**) AGMs stained for Ki-67 expression (brown) are shown. IHC was performed on tissues collected at baseline (preinfection: **i, m**), 150 dpi (**j, n**), 266 dpi (**k, o**) and at necropsy (**l, p**). Images are representative of $n = 6$ AGMs in each group, with at least 9 images taken per animal. On all IHC images, scale bar is 100 μm. ns not significant, sLN superficial lymph nodes.

nonpathogenic SIV infections at the transition from acute-to-chronic infection[40–42]. Therefore, we interrogated whether CD4⁺ T-cell depletion thwarts the control of T-cell immune activation (CD38 and HLA-DR) that normally occurs in AGMs after the acute SIVsab infection. We focused on CD8⁺ T-cells, as they are commonly used to monitor T-cell activation in HIV/SIV infections, and because circulating CD4⁺ T-cells were depleted in the study group.

At baseline, the fractions of activated circulating CD8⁺ T-cells (i.e., HLA-DR⁺ CD38⁺ CD8⁺ T-cells) were similar in both groups (0.40% vs. 0.37%, respectively) (Fig. 7a). The frequency of activated CD8⁺ T-cells

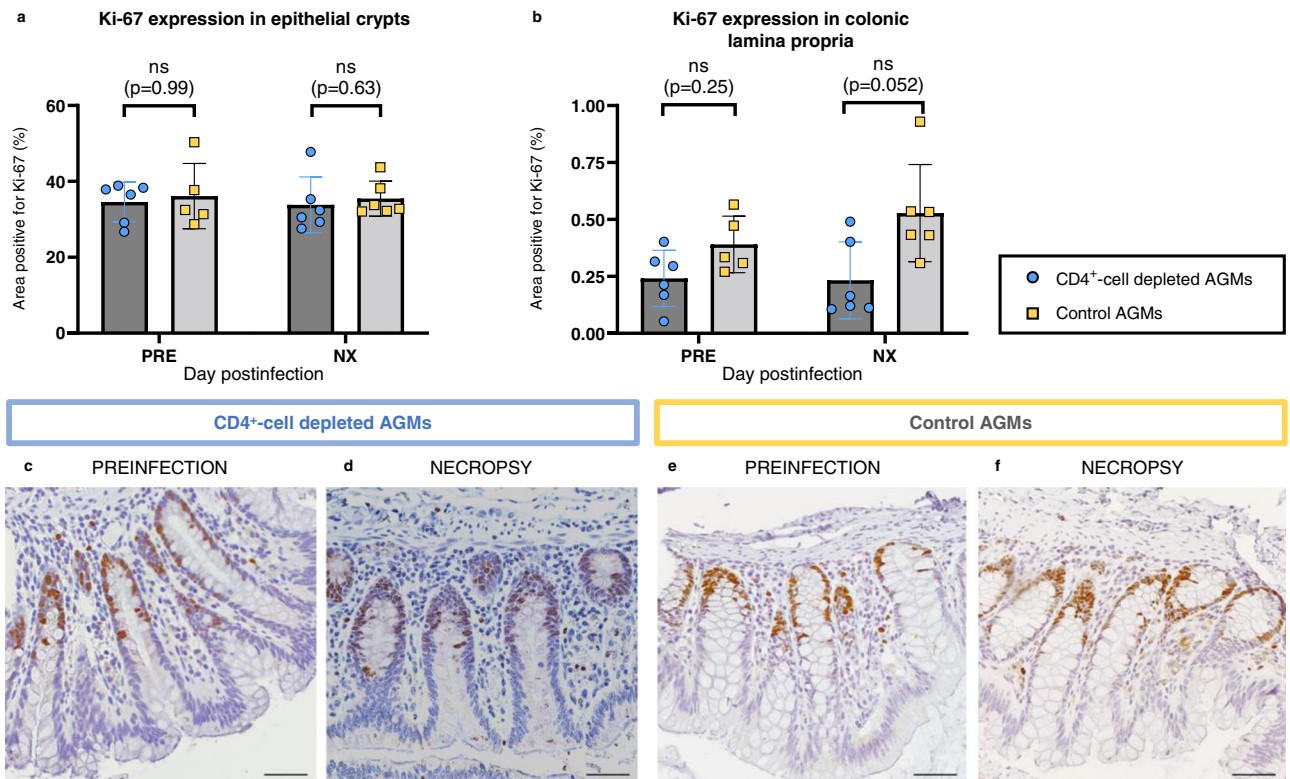

**Fig. 6 | Cell proliferation in the gastrointestinal tract.** Expression of Ki-67 in the epithelial crypts (**a**) and in the lamina propria (**b**) of the colon of CD4$^+$-cell-depleted AGMs (blue circles) and controls (yellow squares). Ki-67 expression was assessed by immunohistochemistry (IHC). Quantifications were performed using the FIJI software. Data presented are means and standard deviations ($n = 6$ AGMs for all, except for controls at baseline, $n = 5$). Data were analyzed with unpaired, two-sided, nonparametric Mann–Whitney tests, followed by Holm-Šídák's correction for multiple comparisons. Representative IHC images of colon tissues of CD4$^+$-cell-depleted (**c**, **d**) and control (**e**, **f**) AGMs stained for Ki-67 expression (brown). Images are representative of $n = 6$ AGMs in each group, with at least 20 images taken per animal. Scale bar: 100 μm. ns not significant.

increased postinfection, reaching a zenith at 10 dpi (2.09% and 1.72%, respectively), followed by a slight decrease at 14 dpi (1.11% vs. 1.30%) and a second peak at 21 dpi (1.42% vs. 1.70%). In both groups, a steady decline occurred after 21 dpi, with T-cell activation reaching baseline levels around 100 dpi (Fig. 7a).

T-cell activation also occurred in the superficial LNs from both groups (Fig. 7b). CD8$^+$ T-cell activation was lower in CD4$^+$-cell-depleted AGMs than in controls at 84 dpi (3.76% vs. 6.74%, respectively), yet this difference was not significant (adjusted $p$ value: 0.22), and the fractions of activated CD8$^+$ T-cells in sLNs were similar in both groups on later time points (Fig. 7b).

In duodenal biopsies, similar dynamics of activated CD8$^+$ T-cells were observed (Fig. 7c), with a 2.5-fold increase between baseline and 21 dpi (from 1.04% to 2.46% in CD4$^+$-cell-depleted AGMs, and from 1.15% to 2.76% in controls). While the fraction of activated CD8$^+$ T-cells in the duodenum started to decline by 42 dpi in controls, it remained elevated in CD4$^+$-cell-depleted AGMs (2.87%), only declining after 73 dpi. After 161 dpi, the fraction of activated CD8$^+$ T-cells in duodenum returned to baseline in both groups.

### CD4$^+$-cell depletion does not alter inflammation
We next assessed the impact of CD4$^+$-cell depletion on acute inflammation and its resolution at the transition to chronic infection.

A slight increase in C-reactive protein (CRP) levels was observed at 10 dpi, with a 1.49-fold and 1.37-fold increase compared to baseline levels for CD4$^+$-cell-depleted AGMs and controls, respectively (Fig. 7d). By 21 dpi, CRP levels returned to baseline, then they rebounded between 28 and 84 dpi, followed by a slow decline thereafter (Fig. 7d). A similar dynamic with 2 peaks during acute infection was observed for CCL5 (RANTES), with no significant differences between the two

groups (Fig. 7e). Other soluble markers of inflammation [CXCL-11 (I-TAC), CXCL-10 (IP-10), CCL-2 (MCP-1), IL-1-RA, CCL-11 (eotaxin) and CXCL9 (MIG)] surged briefly during acute SIV infection, with no significant differences in the magnitude of the peak between the 2 groups (Fig. 7f–k). Most inflammatory cytokines peaked at 10 dpi and returned to baseline levels by 21 dpi, except MCP-1 and MIG which were back to baseline only during chronic infection (Fig. 7f–k). Despite a nearly complete depletion of circulating CD4$^+$ T-cells and a massive loss of mucosal CD4$^+$ T-cells, inflammatory cytokines remained at baseline levels in CD4$^+$-cell-depleted AGMs throughout the follow-up.

### CD4$^+$-cell depletion does not have a discernible impact on GI tract integrity and microbial translocation
Since we observed a severe CD4$^+$ T-cell depletion in different GI tract compartments, and because depletion of specific CD4$^+$-cell subsets involved in the maintenance of epithelial integrity (e.g., Th17 and Th22 CD4$^+$ T-cells) could have a detrimental effect on gut integrity, we next assessed the impact of this persistent CD4$^+$ T-cell depletion on mucosal integrity by IHC assessment of the expression of claudin-3, and lipopolysaccharide (LPS) in colon and LNs, and by quantifying plasma biomarkers: soluble CD163 (sCD163), soluble CD14 (sCD14) and intestinal fatty acid binding protein (I-FABP).

A modest increase in I-FABP was observed by 10 dpi in both groups (1.54- and 1.51-fold increase, respectively) (Fig. 8a). During chronic infection, I-FABP plasma levels were virtually unchanged from baseline for most timepoints, with no significant differences between groups ($p = 0.68$). Staining colon tissues for claudin-3 expression confirmed the integrity of the gut epithelia and the lack of breaks in the epithelium lining (Fig. 8b–f), highlighting that gut integrity was maintained in both groups throughout the follow-up (Fig. 8b–f).

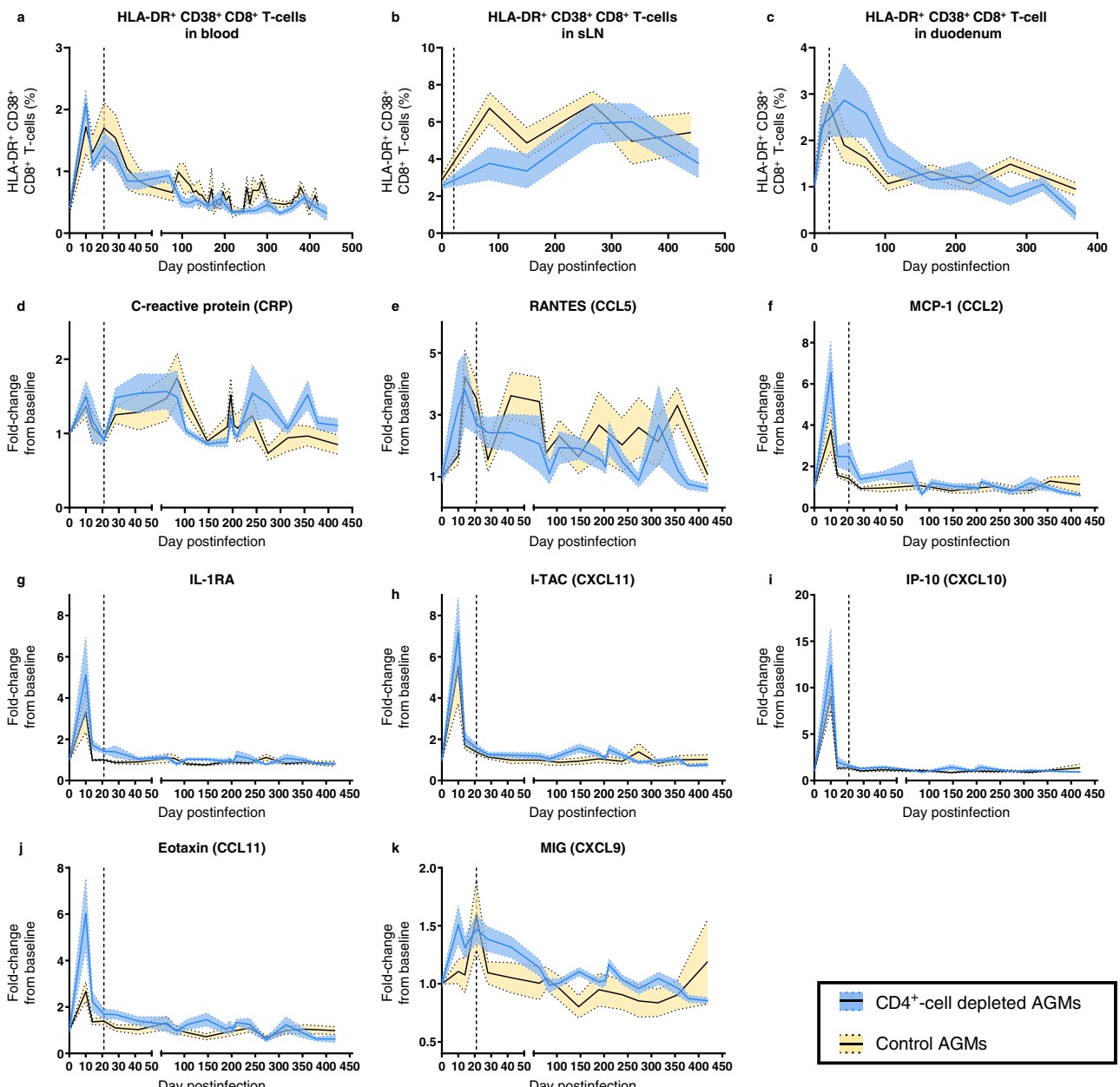

**Fig. 7 | Assessment of immune activation and inflammation.** Immune activation of CD8+ T cells was monitored in circulation (**a**), superficial lymph nodes (**b**) and duodenal biopsies (**c**) of CD4+-cell-depleted AGMs (blue) and controls (yellow), by measuring the percentages of CD8+ T cells expressing both HLA-DR+ and CD38+ by flow cytometry. Systemic inflammation was measured by quantifying C-reactive protein (**d**) and cytokines (**e**–**k**) in the plasma of CD4+-cell-depleted AGMs (blue) and controls (yellow). For all panels, data presented are means (solid lines) and standard errors of the means (shaded regions within the dashed lines), with $n = 6$ AGMs for both groups.

We also monitored the dynamics of different markers of microbial translocation, notably plasma levels of sCD163 and sCD14, as well as LPS levels in colon and sLNs. A modest sCD163 increase occurred during the first 21 dpi, before returning to baseline during chronic infection (Fig. 8g). Plasma sCD14 levels did not significantly change throughout the follow-up (Fig. 8q). Microbial translocation was also evaluated by staining sLNs and colon resections for LPS, and no significant differences could be discerned between the two groups (Fig. 8h–v).

**Impact of CD4+-cell depletion on viral replication**

We next interrogated whether CD4+-cell depletion had any impact on the levels of pVL, as it reduced the availability of target cells in CD4R1-treated AGMs. pVLs peaked at 10 dpi in both groups (Fig. 9a). Mean

pVLs at zenith were 7.5 $\log_{10}$ (range: 7.0–7.7 $\log_{10}$) and 7.4 $\log_{10}$ (range: 6.8–7.8 $\log_{10}$) copies/mL for CD4+-cell-depleted AGMs and controls, respectively (Fig. 9a). Then, pVL declined in both groups (Fig. 9a). In controls, this partial control of viremia persisted throughout the follow-up, with a median pVL of 4.4 $\log_{10}$ copies/mL for timepoints post-50 dpi (IQR: 4.2–4.7 $\log_{10}$ copies/mL). In CD4+-cell-depleted AGMs, viral set-point was 5.5-fold lower ($p = 0.006$), with an average pVL of 3.6 $\log_{10}$ copies/mL after 50 dpi (IQR: 3.0–3.9 $\log_{10}$ copies/mL). Interestingly, at several timepoints, pVLs decreased below the detection threshold for some CD4+-cell-depleted AGMs (Fig. 9b), notably AGM92-18, the AGM with the most profound peripheral CD4+-cell depletion (Supplementary Fig. S6). The CD4+ T-cell depletion did not result in a divergent evolution of the *env* sequences amplified from the plasma of the CD4+-cell-depleted AGMs, compared to the ones

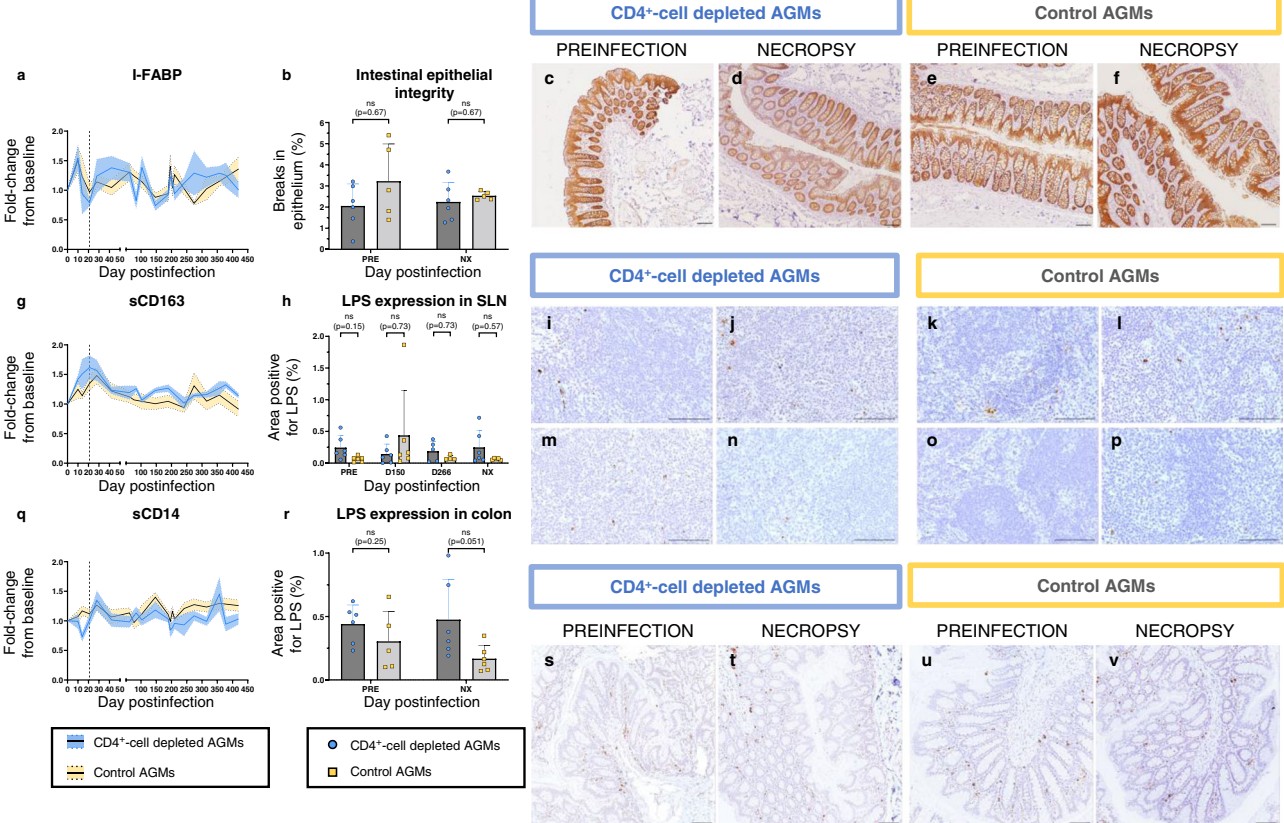

**Fig. 8 | Monitoring of gut integrity and microbial translocation.** Assessment of gut integrity was performed by monitoring a marker of gut damage in plasma, intestinal fatty acid binding protein (I-FABP) (**a**), and the expression of a tight-junction protein, claudin-3 (**b–f**), by immunohistochemistry (IHC) in CD4+-cell-depleted AGMs (blue) and controls (yellow). Assessment of microbial translocation was performed by monitoring two soluble markers of monocyte activation, soluble CD163 (sCD163) (**g**) and soluble CD14 (sCD14) (**q**), that are associated with microbial translocation and disease progression. Microbial translocation was also monitored by quantifying the expression of lipopolysaccharide (LPS, brown) in sLNs (**h–p**) and in colon tissues (**r–v**), by IHC. For IHC, images are representative of $n = 6$ animals in each group, with at least 20 images taken per animal for colon tissues and at least 9 images for sLNs. For all images, the scale bar represents 100 μm. Data presented are means and standard deviations of the means for ELISA (**a**, **g**, **q**), and means and standard deviations for IHC (**b**, **h**, **r**) ($n = 6$ AGMs for CD4+-cell-depleted AGMs and controls, except for (**b**) and (**r**) for which $n = 5$ AGMs for controls at baseline). Data in (**b**), (**h**) and (**r**) were analyzed with unpaired, two-sided, non-parametric Mann–Whitney tests, followed by Holm-Šídák's correction for multiple comparisons. AGMs African Green Monkeys, sLNs superficial lymph nodes, LPS lipopolysaccharide, ns not significant, NX necropsy, PRE preinfection.

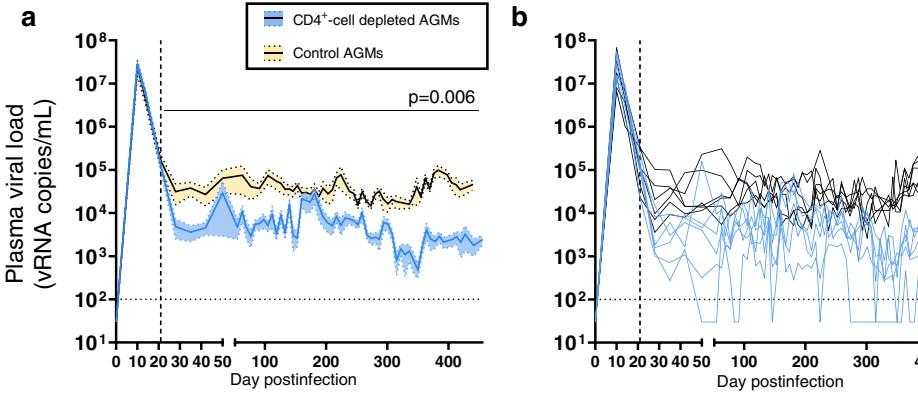

**Fig. 9 | Viral replication in plasma. a** Plasma viral loads (SIVsab92018 RNA copies/mL) were monitored in CD4+-cell-depleted AGMs (blue) and controls (yellow) using a quantitative RT-PCR. Quantification threshold (100 copies/mL) is represented with a dashed black line. Means (solid lines) and standard deviations of the means (shaded regions within the dashed lines) are represented in this panel. Differences between the two groups in the levels of plasma viral loads after administering the CD4-depleting antibody were assessed using mixed-effects models, with Holm-Šídák's correction for multiple comparisons. Mixed-effects model analysis revealed that CD4+-cell depletion had a significant effect on the levels of plasma viral loads, and the corresponding p value is reported in the top right corner. **b** Individual plasma viral loads in CD4+-cell-depleted (blue) and control (black) AGMs. Data shown in (**a**) and (**b**) are the averages of 2 PCR for all samples, with $n = 6$ AGMs for both groups.

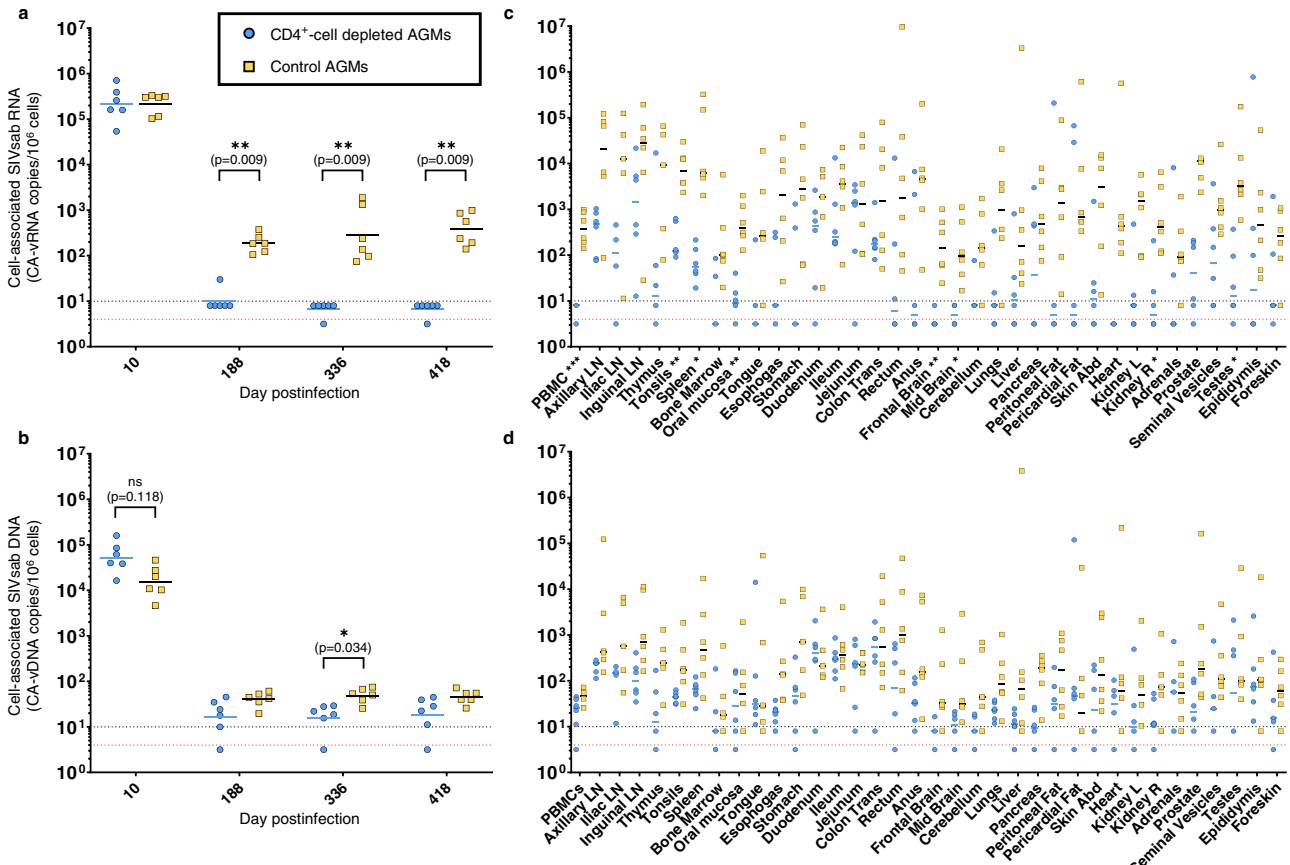

**Fig. 10 | Viral reservoirs in tissues at necropsy.** Viral reservoirs were assessed in peripheral blood mononuclear cells (PBMC) at different timepoints (**a**, **b**) and in multiple tissues at necropsy (**c**, **d**). Levels of cell-associated viral RNA (CA-vRNA: **a**, **c**) and DNA (CA-vDNA: **b**, **d**) were measured in CD4+-cell-depleted AGMs (blue circles) and controls (yellow squares) using quantitative RT-PCR (for CA-vRNA) or PCR (for CA-vDNA), and they are expressed as SIVsab92018 RNA or DNA copies per $10^6$ cells. An arbitrary value of 10 copies/$10^6$ cells was attributed to samples in which SIV RNA or DNA was detected but was below the quantification threshold (10 copies/$10^6$ cells, black dashed line). Similarly, an arbitrary value of 5 copies/$10^6$ cells was attributed to samples in which SIV RNA or DNA was not detected (red dashed line). Data shown in panels A-D are the averages of 2 PCR, with $n = 6$ AGMs for both groups, except for some tissues at necropsy due to tissue availability (for CD4+-cell depleted AGMs: $n = 4$ for iliac lymph nodes, $n = 5$ for cerebellum, adrenals and foreskin; for controls: $n = 5$ for iliac lymph nodes, thymus and duodenum). Data were analyzed using unpaired, two-sided, nonparametric Mann–Whitney tests, followed by Holm-Šídák's correction for multiple comparisons (**a**, **b**) or with unpaired $t$ tests with Welch correction, followed by Holm-Šídák's correction for multiple comparisons (**c**, **d**). Asterisks indicate statistical significance. Abbreviations: *$p < 0.05$; **$p < 0.01$; ***$p < 0.001$.

from controls (Supplementary Fig. S7A). The mean genetic distance between the parental strain, SIVsab92018, and the *env* sequences were similar between the 2 groups, at 21 dpi (i.e., before initiating the CD4+-cell-depleting antibody treatment) and at time of necropsy (Supplementary Fig. S7B). Furthermore, when visually inspecting the *env* sequences, CD4+-cell-depleted AGMs did not exhibit any change in the functional sites associated with the emergence of CD4-independent envelopes[43] (loss of asparagine-linked glycosylation sites), or with a switch from a R5 to a X4 tropism[44] (insertions and/or change in the net charge of the V3 loop) (Supplementary Fig. S7C).

### Impact of CD4+-cell depletion on reservoir dynamics

Viral reservoirs can be assessed by monitoring cell-associated viral RNA and DNA (CA-vRNA and CA-vDNA, respectively). At 10 dpi, CA-vRNA levels in PBMC were similar in both groups, with 5.3 and 5.3 log$_{10}$ RNA copies/$10^6$ PBMCs in average for CD4+-cell-depleted AGMs and controls, respectively (Fig. 10a). After CD4+ T-cell depletion, CA-vRNA were undetectable in all tested samples but one (Fig. 10a). CA-vDNA levels in PBMC were also lower than in controls, but the differences were only significant at 188 dpi (Fig. 10b). Both CA-vRNA and CA-vDNA levels were assessed in multiple tissues at necropsy. Median CA-vRNA levels were significantly lower in PBMC, tonsils, oral mucosa, testes, spleen, brain (frontal and midbrain) and right kidney of CD4+-cell-

depleted AGMs (Fig. 10c). Similarly, CA-vDNA levels were lower in CD4+-cell-depleted AGMs than in controls, although these differences did not reach statistical significance for any tissue (Fig. 10d).

## Discussion

Persistent low CD4+ T-cell counts in blood and lymphoid tissues are one of the hallmarks of progressive HIV/SIV infections[1], CD4+ T-cell counts being positively correlated with disease progression[20,45]. In pathogenic infections, CD4+ T-cell depletion occurs early and persists throughout the infection, despite an increased CD4+ T-cell proliferation[46]. Eventually, this continuous CD4+ T-cell loss leads to a severe impairment of the CD4+ T-cell homeostasis, which precedes progression to AIDS by a few years[47].

There are several unanswered questions regarding the key determinants of HIV/SIV disease progression. Notably, the relative contributions of CD4+-cell depletion *vs.* GI tract dysfunction in driving HIV/SIV disease progression are not clearly defined, and the potential role of CD4+ T-cell loss in inducing HIV/SIV-associated GI tract dysfunction is not elucidated. The main reason is that gut dysfunction and severe depletion of CD4+ T-cells are intricately associated in HIV/SIV infections, as they occur almost simultaneously early in infection.

In African NHPs that are natural hosts of SIV, retroviral infections are nonpathogenic, i.e., animals do not progress to AIDS, despite a very

robust viral replication and a massive mucosal CD4[+] T-cell depletion during acute infection[30,48–51]. During chronic infection, they restore circulating CD4[+] T-cells to baseline levels and present a partial restoration of mucosal CD4[+] T-cells[30]. This is not due to enhanced CD4[+] T-cell homeostasis in these species, as CD4[+] T-cell restoration rates after experimental depletion did not differ between uninfected macaques and sooty mangabeys (i.e., natural and non-natural hosts of SIV)[52]. Natural hosts have developed effective mechanisms to maintain their GI integrity throughout SIV infection[33,34], enabling them to control excessive T-cell activation and systemic inflammation at the transition from acute-to-chronic infection[53], and to recover their CD4[+] T-cells during the chronic infection.

Altogether, these features suggest that the transient acute depletion of CD4[+] T cells is not sufficient to cause disease progression[30], and that the ability of the African NHP hosts to avert disease progression is linked to their unique ability to maintain GI tract integrity throughout infection[33]. Natural hosts of SIV are thus an ideal model to decipher the relative contribution of GI tract dysfunction and CD4[+] T-cell depletion on the outcomes of SIV infection, as those parameters can be separately manipulated.

Here, we aimed to decipher the role of CD4[+] T-cell depletion in driving the HIV/SIV-associated gut dysfunction and progression to AIDS, by disrupting immune CD4[+] T-cell homeostasis for over a year through repeated infusions of an anti-CD4 mAb to SIVsab-infected AGMs, a model of nonpathogenic SIV infection. Initiating the experimental CD4[+]–cell depletion after the peak of SIVsab replication allowed us to assess whether, in the context of a steady viral replication, the sole depletion of CD4[+] T-cells induced gut damage and/or caused disease progression.

We successfully depleted virtually all circulating CD4[+] T-cells and over 90% of the CD4[+] T-cells in the GI tract for over a year. CD4[+] T-cell restoration was extremely limited between infusions, as previously reported[52]. One strength of our study is the magnitude of the achieved CD4[+] T-cell depletion in tissues, at least equivalent to that observed in humans and NHPs that progress to AIDS[4]. Previous studies of experimental CD4[+] T-cell depletion using CD4-depleting mAb, which were mainly performed on RMs and for shorter periods of time, achieved a massive depletion of circulating CD4[+] T-cells, but efficacy in tissues was more limited (around 50 to 60% depletion in the GI tract)[27–29,54]. As only 2% of the total CD4[+] T-cell population is in circulation and the GI tract is the main reservoir of CD4[+] T-cells[55], particularly the CD4[+] T-cell subsets targeted by HIV/SIV (activated, CCR5-expressing, memory CD4[+] T-cells)[1], a limited CD4[+] T-cell depletion at mucosal sites translates into a large pool of residual CD4[+] T-cells, which can alter the results of such experiments. However, it must be noted that the levels of CD4[+] T-cells are significantly lower in uninfected AGMs than RMs[30,56], which could be a key contributor to the observed profound depletion of mucosal CD4[+] T-cells.

An indirect confirmation of the efficacy of our approach to massively reduce CD4[+] T-cells is the lower viral replication in the CD4[+]-cell-depleted AGMs, probably resulting from a significant reduction of the target cells[28]. This reduced viral replication was observed both in circulation and in tissues, although these differences reached statistical significance only for a limited number of tissues. Recently, a modest reduction of viral reservoirs after antibody-mediated CD4[+] T-cell depletion has been reported in SIVmac239-infected, ART-treated RMs[29]. The reduction in the size of the viral reservoir was probably more prominent in AGMs because ART was initiated at 4 dpi in RMs, limiting the seeding of the viral reservoir, while the AGMs were not subjected to any intervention that could have alter the reservoir seeding. To note, the depletion of CD4-expressing cells did not induce a divergent evolution of SIVsab in AGMs. No significant changes in the *env* sequences were observed in CD4[+]-cell-depleted AGMs, suggesting that the SIVsab strains did not evolve to be CD4-independent despite the paucity of target cells.

Throughout the follow-up, no significant gut dysfunction was observed in the CD4[+]-cell-depleted AGMs, none of which progressed towards AIDS, nor developed opportunistic infections. One limitation of our study is that the lower viral replication in CD4[+]-cell-depleted AGMs could have impacted disease progression. However, viral replication is not the sole factor contributing to disease progression, as progression to AIDS was reported to occur in PWH with low to undetectable viremia[57], in which increased immune activation preceded disease progression. No increase in markers of gut dysfunction (sCD14, IFABP, sCD163, LPS levels in tissues) or inflammation (CRP, sCD163) was seen in CD4[+]-cell-depleted AGMs during the follow-up. This suggests that prolonged CD4[+] T-cell depletion is not fueling SIV-associated gut dysfunction, nor driving SIV disease progression. Furthermore, disease progression is usually occurring swiftly in NHPs, i.e., within a year[50]. Hence, it is unlikely that our AGMs would have progressed to AIDS should the follow-up been extended beyond the 400 days. Together, these observations suggest that a profound and persistent CD4[+] T-cell depletion can be tolerated without any major side effect and does not induce disease progression, when epithelial gut integrity is preserved, and chronic T-cell activation and systemic inflammation are kept at bay. However, other mechanisms might have been involved in AGMs' protection against disease progression, including the ability to mount a more efficient acute type I IFN response, and that that CD4[+] T-cell depletion could have been insufficient to revert their ability to control SIV infection. The role of this efficient immune response of natural hosts of SIV could not be assessed in this study, as we initiated the CD4-depleting antibody treatment at 21 dpi, i.e., after the innate immune response, to avoid altering virus reservoir seeding during acute infection. Lymphopoiesis could also be differently altered in our model of experimental CD4[+] T-cell depletion, compared to HIV/SIV progressive infections, and it has been previously shown that exhaustion of lymphopoiesis can drive disease progression[58,59]. Furthermore, our model might not recapitulate other characteristics of pathogenic HIV/SIV infections, in which multiple immune cell types are dysfunctional due to the persistent immune activation and inflammation that persist even during ART.

In previous studies, we have reported that immune cell subsets other than CD4[+] T-cells [i.e., CD4[neg] CD8[neg] double negative T-cells[60] and CD8αα T-cells[37]] can exert some CD4-like immune functions in AGMs, as an adaptive protection mechanism aimed at maintaining the homeostasis of the helper function and avoiding progression to AIDS in the context of a massive CD4[+] T-cell lymphopenia. One mechanism by which AGMs have coevolved with SIVsab is the downregulation of CD4 by CD4[+] T-cells via epigenetic silencing of the CD4 gene[61]. Therefore, we assessed both the counts and functionality of these immune cell subsets. We did not observe any significant increase in counts or functionality of CD8αα T-cells or double negative T-cells in CD4[+]-cell-depleted AGMs, compared to controls. Thus, CD4[+]-cell depletion does not induce an expansion of those cell subsets, or an increase in the percentages of those cells exerting CD4-like immune functions. However, it is also possible that the sole presence of those cells in AGMs is sufficient to compensate the loss of CD4[+] T-cells by ensuring part of their immunological functions, thus contributing to the protection of AGMs against AIDS.

Overall, our results suggest that, even though restoring CD4[+] T-cell counts in blood and tissues are important objectives of ART, it might not be necessary to reach preinfection levels to avert disease progression. While multiple mechanisms could contribute to AGM protection against disease progression, we showed that damages to the gut integrity in SIV-infected AGMs resulted in an increase in markers associated with disease progression[62]. This suggests that gut damage can override the effects of other mechanisms preventing disease progression in the natural hosts of SIVs, and that preservation of gut integrity could benefit to humans too[62]. Here, as epithelial gut integrity was maintained, there was no gut dysfunction, microbial

translocation was averted, and local and systemic inflammation and immune activation were not increasing. When those conditions are met, NHPs do not progress to AIDS, regardless of the degree of CD4+ T-cell depletion and despite a robust viral replication. This might also be true for persons living with HIV, as not all individuals progress to AIDS when reaching a set number of circulating CD4+ T-cells[45]. It has been notably shown that persons living with HIV-2 progress to AIDS with higher CD4+ T-cell counts than persons living with HIV-1[63]. On the opposite side of the spectrum, low CD4+ T-cell counts are not systematically associated with opportunistic infections. Although most patients with idiopathic CD4 lymphopenia do present opportunistic infections, it is not systematic, especially for patients with autoantibodies[64]. Not all the PWH present opportunistic infections at the same levels of circulating CD4+ T-cells. It can be hypothesized that some PWH could have a delayed disease progression, due to a better preservation of their gut barrier, and subsequent limited inflammation and immune activation. This is encouraging for clinical care, as restoring mucosal CD4+ T-cell counts to levels close to those of uninfected individuals is complicated, especially when ART is not initiated early[17]. While our results strongly suggest that prolonged disruption of the gut integrity, and not CD4+ T-cell depletion, is the main driver of disease progression, further studies should aim at assessing the outcomes of a nonpathogenic retroviral infection in the context of a damaged gut barrier and evaluate therapeutic strategies aimed at restoring epithelial gut barrier during chronic infections.

## Methods

### Animal model
Twelve male AGMs (*Chlorocebus sabaeus*) were included in this study. AGMs were fed and housed according to regulations set forth by the Guide for the Care and Use of Laboratory Animals and the Animal Welfare Act[65]. All AGMs included in this study were social housed (paired) in stainless steel cages, had 12/12 light cycle, were fed twice daily with regular chow (Monkey Diet 5038, LabDiet, St Louis, MO, USA), and water was provided ad libitum. A variety of environmental enrichment strategies were employed. Furthermore, AGMs were observed twice daily, and any signs of disease or discomfort were reported to veterinarians for evaluation. For sample collection, AGMs were anesthetized with 10 mg/kg ketamine HCl (Park-Davis, Morris Plains, NJ, USA) or 0.7 mg/kg tiletamine HCl and zolazepam (Telazol, Fort Dodge Animal Health, Fort Dodge, IA) injected intramuscularly. They were euthanized by intravenous (iv) administration of barbiturates, prior to the onset of any clinical signs of disease.

### Study approval
All AGMs were housed at the University of Pittsburgh, according to the Association and Accreditation of Laboratory Animal Care (AAALAC) guidelines. Animal experiments were approved by the University of Pittsburgh Institutional Animal Care and Use Committee (IACUC) (protocol #19074902).

### Virus
All AGMs were iv infected with plasma equivalent of 300 tissue culture infectious doses (TCID$_{50}$) of SIVsab92018[41].

### Samples
Duodenal and colon biopsies were collected prior to infection, as described[30,66]. During acute infection, duodenal biopsies were taken at 10 and 21 dpi. During chronic infection, an intestinal biopsy was taken every month, alternating between duodenal and colon biopsies. Superficial lymph nodes (LNs) were collected prior to infection, then approximately every three months during infection and at necropsy.

Plasma was separated from whole blood by centrifugation (1500 × g for 20 min), and peripheral blood mononuclear cells (PBMCs) were isolated using a Ficoll density gradient centrifugation.

Intestinal biopsies were washed with EDTA for 20 min at 37 °C, then subjected to collagenase digestion at 37 °C with agitation. Cell suspension was filtered through a 70 μm filter, then layered in a tube containing 2 mL of a 35% Percoll solution on top of 2 mL of a 60% Percoll solution. After a centrifugation at 1500 × g for 20 min, lamina propria lymphocytes (LPL) were retrieved at the interphase between the two Percoll solutions. LNs were mechanically minced, pressed through a 70 μm nylon mesh screen, filtered through 70 μm nylon mesh bags and washed with RPMI medium (Cellgro, Manassas, VA) containing 5% heat-inactivated newborn calf serum, 0.01% penicillin-streptomycin, 0.01% L-glutamine and 0.01% HEPES buffer. Freshly isolated cells were then used for flow cytometry.

### Antibodies and flow cytometry
White blood cells from blood and mononuclear cells isolated from intestinal biopsies and LNs were immunophenotyped by flow cytometry, as described[67]. First, TruCount staining was performed on 50 μL of whole blood, using CD45 (PerCP-Cy5-5, 3 μL, clone D058-1283), CD3 (V450, 2 μL, SP34-2), CD14 (PE-Cy7, 4 μL, M5E2), CD16 (APC-Cy7, 2 μL, 3G8) and CD163 (APC, 5 μL, GHI/61) antibodies. This allowed us to precisely quantify circulating CD45+ cells, T cells and monocytes. The exact counts of CD4+ and CD8+ T-cells were determined by multiplying the exact number of CD3+ T-cells by the percentage of CD4+ or CD8+ cells among CD3+ T-cells that were determined in a different staining. CD14 and CD16 were used to identify circulating monocyte subsets. Whole peripheral blood (100 μL) and cells isolated from tissues were stained with fluorescently-labeled antibodies (all purchased from BD Bioscience, San Jose, CA, USA, unless noted otherwise): Annexin V (5 μL, FITC), CCR5 (10 μL, PE), CD3 (V450, 2 μL, clone SP34-2), CD4 (APC, 2.5 μL, L200), CD8 (PE-CF594, 3 μL, RPA-T8), CD14 (PE-Cy7, 4 μL, M5E2), CD16 (APC-Cy7, 4 μL, 3G8), CD163 (APC, 5 μL, GHI/61), CD20 (APC-H7, 4 μL, 2H7), CD28 (PE-Cy7, 3 μL, CD28.2), CD38 (FITC, 15 μL, AT-1) (Stemcell), CD69 (APC-H7, 5 μL, FN50), CD95 (FITC, 15 μL, DX2), HLA-DR (PE-Cy7, 3 μL, L243), Ki-67 (FITC or PE, 20 μL, B56), Live/Dead Aqua (200 μL of a 1:500 dilution, Thermofisher). For Ki-67 staining, cells were fixed, permeabilized with 1X BD Fix/Perm, then stained for Ki-67. All cells were washed then fixed with BD Fix before being analyzed. Flow cytometry acquisitions were performed on an LSR II or a LSR Fortessa flow cytometer (BD Biosciences). Flow data were analyzed using FlowJo® v10.8.0 (TreeStar). An example of the gating strategy is detailed in Supplementary Fig. S8.

### Intracellular cytokine staining
Cryopreserved cells were thawed, washed, and split into two tubes, each containing approximately 2 × 10$^6$ cells in 1 mL of complete RPMI media. Cells in one tube were stimulated with 5 ng/ml PMA and 1 μg/ml ionomycin. Both tubes were treated with 10 ng/ml of brefeldin A, 1 μl/ml monensin (BD GolgiStop), and CD107a (BV711, 5 μL, clone H4A3) for 16–18 h at 37 °C. Cells were washed twice in PBS and stained with fixable Live/Dead Aqua (1 μL), CD3 (ALX700, 3 μL, SP34-2), CD4 (BV605, 5 μL, L200), CD8 (PerCP-Cy5.5, 5 μL, RPA-T8), CD28 (ECD, 5 μL, CD28.2) and CD95 (Cy5PE, 1 μL, DX2) for 20 min at 4 °C. Cells were washed with PBS and permeabilized with Foxp3 Perm Solution (eBioscience) for 1 h at 4 °C. Cells were washed twice with 2 mL FoxP3 Perm Wash Buffer Solution (eBioscience) and stained with FoxP3 (Alx488, 5 μL, 206D), GranzymeB (PE, 0.25 μL, GB11), IFNγ (Cy7PE, 0.5 μL, 4 S.B3), IL-2 (BV785, 5 μL, MQ1-17H12), IL-17 (e450, 5 μL, eBio64DEC17), and CD40L (APC-e780, 5 μL, 24–31) for 30 min at 4 °C. Cells were washed twice with 2 mL FoxP3 Perm Wash Buffer Solution and fixed in 1% PFA before being analyzed on a LSR Fortessa. Unstimulated cells were used to set gates for stimulated cells from the same animal and timepoint to account for background staining. Data was analyzed using FlowJo® v10.7.1, Graphpad Prism® v9.3.1 (GraphPad), and SPICE 6. Data was only included for analysis if at least 200 cells were present in the target population.

## Plasma viral loads

Viral RNA (vRNA) were extracted from plasma using QIAGEN viral RNA Mini kit (QIAGEN, Germantown, MD). A quantitative reverse-transcription PCR (qRT-PCR) with a limit of quantification of 100 copies/mL was performed as described[41], using the following set of primers (SIVsab-F: 5′-CAGCCCTGCTGAAACAATATG-3′, SIVsab-R: 5′-GC TTTAAGCCTCCTCGTAAG-3′) and probe (SIVsab-Probe: 5′-/56-FAM/ CAGAGGCAA/ZEN/AGCAGACAGAGCAGT/3IABkFQ/−3′). All RT-PCR were performed in duplicate, on a 7900HT Fast Real Time System (Applied Biosystems).

## Cell-associated (CA) vRNA and vDNA

Nucleic acids were extracted from snap-frozen tissue sections using a TRIzol-based protocol. Briefly, 400 μL Tri-reagent (Molecular Research Center, Inc, Cincinnati, OH) was added to each tube before vortexing them using a SPEX Geno/Grinder (SPEX SamplePrep, Metuchan, NJ). After tissue homogenization was complete, 100 μL of 1-bromo-3-chloropropane (BCP) (Molecular Research Center, Inc, Cincinnati, OH) was added to 1 mL of tissue lysate, then samples were vortexed for 15 s and centrifuged at 14,000 × $g$ for 15 min at 4 °C. The upper aqueous RNA phase was transferred into a tube containing 12 μL of 20 mg/mL glycogen. After mixing, 500 μL of isopropanol were added, mixed then centrifuged at 21,000 × $g$ for 10 min at room temperature. Isopropanol was discarded and the pellet was washed with 70% ethanol and centrifuged for 5 min, supernatant was discarded, and the pellet was air-dried for 10 min, then resuspended in molecular grade (DNAse, RNAse free) water. DNA phase was also extracted, using 500 μL DNA Back Extraction solution (4 M GuSCN, 1 M Tris base, 50 mM sodium citrate) instead of BCP. Extracted RNA and DNA samples were subject to the same qRT-PCR as for VL quantification, except the reverse transcription step was omitted for DNA samples. DNA extracts were used to quantify the number of cells present in each sample by amplifying CCR5 DNA[33], with primers (CCR5-F: 5′-CCA GAAGAGCTGCGACATCC-3′, CCR5-R: 5′-GTTAAGGCTTTTACTCATCT CAGAAGCTAAC-3′) and probe (CCR5-Probe: 5′-/56-FAM/TTCCCCTAC/ ZEN/AAGAAACTCTCCCCGGTAAGTA/3IABkFQ/−3′). The final calculated number of viral RNA and DNA copies were divided by the number of cells present in the sample to determine the number of CA-vRNA and CA-vDNA copies per million cells[68]. All PCRs were performed in duplicate.

## ELISA and luminex assays

Different soluble markers were quantified in plasma of AGMs, using immuno-assays: sCD14 (Quantikine Human sCD14 Immunoassay, R&D Systems, Minneapolis, MN, USA), sCD163 (Macro163, IQProducts, Netherlands), I-FABP (Monkey I-FABP/FABP2 ELISA Kit, MyBioSource, San Diego, CA, USA) and CRP (Monkey CRP ELISA kit, Life Diagnostics, PA, USA). Expression of cytokines and chemokines was quantified using a magnetic bead-based assay, the Cytokine 29-plex Monkey Panel (Thermofisher), and plates were read on a MAGPIX® instrument. All ELISA and Luminex assays were performed according to manufacturer's instructions. Due to the relatively small sample size and the relatively large interindividual variation for some analytes, the levels of all markers are expressed as a fold-change compared to preinfection levels that were determined on 2−3 preinfection samples for all AGMs.

## Chemistry panels

Comprehensive chemistry panels were performed on undiluted sera of AGMs by IDEXX® BioAnalytics laboratories (North Grafton, MA, USA).

## Immunohistochemistry (IHC)

IHC analysis of Ki-67, claudin, and lipopolysaccharide (LPS) were performed on formalin-fixed, paraffin-embedded tissues. Four μm-thick sections were deparaffinized, rehydrated, and rinsed. For antigen retrieval, the sections were microwaved in Vector Unmasking Solution (Vector Laboratories Burlingame, CA, USA) and treated with 3% hydrogen peroxide. Sections were incubated with 200 μL of a primary antibody at a 1:100 dilution. The following primary antibodies were used: Ki-67 (mouse monoclonal antibody, clone MIB-1, Dako), claudin-3 (rabbit polyclonal antibody, RB-9251-P1, Thermo Fisher), LPS (mouse monoclonal antibody, clone WN1 222-5, Hycult Biotech, USA). Secondary antibodies and avidin/biotin complex used were from the Vector Vectastain ABC Elite Kits (VectorLabs, Burlingame, CA, USA). For visualization, sections were treated with DAB (Dako), counterstained with hematoxylin, dehydrated, and mounted in a xylene-based mounting media. Images were taken on a Nikon Ni-U microscope. Quantification was performed using the FIJI image software, as described[33], with at least 20 images per sample for intestinal tissues and liver, and at least 9 images for sLN (magnification x400). Positive signal was isolated via color threshold; percent area positive for the marker was measured for each image and the mean was calculated.

## Genetic analyses

Full SIVsab $env$ sequences were amplified from plasma at 21 dpi for all AGMs (6 CD4[+]-cell-depleted and 6 controls) and at time of necropsy in 9 AGMs (5 CD4[+]-cell-depleted and 4 controls), sequenced as described[69], and deposited in GenBank (accession numbers: OP491432-OP491452). A dataset composed of 36 SIVsab $env$ sequences (21 obtained in this study, plus 15 retrieved from the Los Alamos National Laboratory website) was prepared. The phylogenetic tree was built with PhyML 3.0, using a maximum likelihood method with a General Time Reversible (GTR) model, with 500 bootstraps[70]. Evolutionary divergence was also estimated by computing the genetic distance (i.e., the number of base substitutions per site) between each $env$ sequence and the reference strain, SIVsab92018. Analyses were conducted using the Maximum Composite Likelihood model in MEGA11[71]. Finally, $env$ sequences were visually inspected for mutations and/or insertions and deletions in the potential asparagine-linked glycosylations sites (N-X-S/T motifs) located in the V1, V2 and V5 loops, and in the entire V3 loop sequence. The net charge (sum of positively charged amino acids minus the sum of negatively charged amino acids) of the V3 loop was also calculated.

## Statistics and reproductivity

No formal sample size determination was performed. The size of the groups was based on historical data and knowledge. No data were excluded from the analyses, all assays were performed on the twelve AGMs at all timepoints. For some experiments, assays could not be performed for all animals, due to the lack of tissue or to an insufficient number of cells to perform flow cytometry experiments. The investigators were not blinded to allocation during experiments and outcome assessment (except for PCR assays), as they were involved in cell separation and preparation of the dilutions of the CD4-depleting antibody. All statistical analyses were performed using GraphPad Prism v9.3.1. For all tests, $p$ values below 0.05 (95% Confidence Interval) were considered statistically significant. For immune cells, pVLs and biomarkers, when baseline levels were similar, differences of means between the 2 groups for samples collected after 21 dpi - i.e., after the first administration of CD4R1- were assessed using mixed-effects models, to account for repeated measures, followed by Holm-Šídák's correction for multiple comparisons. IHC data and cell-associated RNA and DNA levels were analyzed with unpaired, two-sided, nonparametric Mann−Whitney tests, followed by Holm-Šídák's correction for multiple comparisons. For viral reservoirs in tissues collected at necropsy and percentage of CD4[+] T-cells in different tissues at necropsy, unpaired, two-sided $t$ tests with Welch correction were performed, followed by Holm-Šídák's correction for multiple comparisons. To assess the evolution of functional profiles of lymphocyte populations, mixed-effects models were used, with Dunnett's correction for multiple comparisons.

**Reporting summary**

Further information on research design is available in the Nature Portfolio Reporting Summary linked to this article.

## Data availability

Source data files are provided with this paper (https://doi.org/10.6084/m9.figshare.21200539.v1). All SIVsab env sequences are deposited in GenBank database (accession numbers: OP491432-OP491452).

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

## Acknowledgements

The anti-CD4 antibody (CD4R1) used in this study was provided by the NIH Nonhuman Primate Reagent Resource (P40 OD028116). This work was supported by grants from the National Institutes of Health/National Institute of Diabetes and Digestive and Kidney Diseases/National Heart, Lung and Blood Institute/National Institute of Allergy and Infectious Diseases: R01 DK130481 (IP), R01 DK113919 (IP/CA), R01 DK119936 (CA), R01 DK131476 (CA), R01 HL117715 (IP), R01 HL123096 (IP), R01 HL154862 (IP), R01 AI119346 (CA). This study was funded, in part, by the Division of Intramural Research/NIAID/NIH and NHLBI/NIH. The content of this publication does not necessarily reflect the views or policies of the Department of Health and Human Services, nor does mention of trade names, commercial products, or organizations imply endorsement by the U.S. Government. The funders had no role in study design, data collection and analysis, decision to publish, or preparation of the manuscript.

## Author contributions

Conceptualization: Q.L.H., P.S., E.B.C., J.D.E., J.M.B., C.A. and I.V.P.; Methodology: Q.L.H., P.S., C.X., A.R.R., J.M.B., C.A., I.V.P.; Experiments: Q.L.H., P.S., C.X., A.R.R., L.T., H.A., A.K., E.B.C., R.S., S.S., T.H., D.J.C., D.M.; Data analysis: Q.L.H., P.S., A.R.R., L.T., E.B.C., J.M.B., C.A. and I.V.P.; Supervision: J.D.E., J.M.B., C.A. and I.V.P.; Funding acquisition: C.A. and I.V.P.; Writing—original draft: Q.L.H., A.R.R., J.M.B., C.A. and I.V.P. All the authors reviewed and edited the manuscript.

## Competing interests

The authors declare no competing interests.
