## [Peer Review File · Nature Communications]

Prolonged Experimental CD4+ T-Cell Depletion Does Not Cause Disease Progression in SIV-infected African Green MonkeysREVIEWER COMMENTS

Reviewer #1 (HIV, SIV, NHP models) (Remarks to the Author):

The authors have been working on the exact mechanism of AIDS progression in HIV/SIV infections. In particular, they have been working on a non-pathogenic model of SIV infection in African Green Monkeys (AGM) and a pathogenic model of SIV infection in Rhesus Monkeys (RM) (ref. 29-31). They have accumulated data indicating that gastrointestinal (GI) tract maintenance is the key for not developing AIDS in SIV-infected AGMs, while CD4 depletion could not be the main driver for AIDS progression. For this conclusion, further evidence should be accumulated. This study showing non-acceleration of AIDS progression by prolonged antibody-mediated CD4+ T cell depletion in SIV-infected AGMs adds an important data supporting the above conclusion, although unanswered questions still remain. Results on the prolonged antibody-mediated CD4+ T cell depletion are valuable. The following issues should be addressed.

1. One of the limitations of this study is reduction in viral loads (VL) by CD4 depletion, possibly by loss of SIV target cells. VL reduction could mitigate the impact of CD4 depletion itself on disease progression. Antibody-mediated CD4 depletion may be different from CD4 depletion under viral replication and CD4 restoration from acute CD4 depletion in GI. The authors should mention this point more clearly with more detailed discussion.
2. Related with the comment 1, VL reduction may result in reduction of immune activation. However, difference in immune activation between CD4-depleted and Controls was unclear. In Fig 4, CD8ab+IFN γ + (& CD8abGrzB+) was increased on d371 in CD4-depleted. Some discussion is required on these points.
3. It remains unclear whether prolonged antibody-mediated CD4+ T cell depletion accelerates AIDS progression in SIV-infected RMs. In addition, previous reports showed difference in type I IFN responses in SIV infection between AGMs and RMs. It is recommended to have a discussion on these points.
4. In Fig 2B, individual means of Controls is expected to be 100, but data shown are not. Please explain the reason.
5. CD4 depletion may result in change in viral tropism. It would be better to show data on env sequences and/or viral tropisms.
6. Ref 28 should be corrected (lacking in page number?).

Reviewer #2 (HIV therapy, SIV models) (Remarks to the Author):

Breakdown of the gut mucosal integrity is one of the hallmarks of HIV infection. Several factors may be involved in this intestinal damage. Central amongst them is the profound loss of mucosal CD4+ T cells, that have key role in the control of systemic breakthrough by enteral bacteria. African green monkeys are natural hosts of SIV_{gab}. They do not show signs of intestinal damage even in the presence of robust SIV proliferation. Le Hingrat et al. investigated if prolonged CD4+ T cell depletion would elicit breakdown of the gut mucosal integrity in a cohort of SIV_{gab}-infected African Green Monkeys. Although the depletion was nearly 90% in the intestine, (and even higher in other tissues and the blood) they were not able to find any sign of deterioration of intestinal integrity, chronic activation of the immune system, or functional takeover of the role of CD4 T cells by remaining T cell populations. The significance of this finding is that we may not need the total restoration of the mucosal CD4 T cell pool to maintain the integrity of the intestinal barrier. Perhaps there are other factors that could be targeted by novel therapeutics. However, the questions still remain:

- is 90% depletion leaves enough CD4 T pool in critical mucosal compartments that one year follow-up is not enough to achieve the detrimental effect,
- can the mechanism that renders the African green monkeys refractory against the disease progression be found in humans as well?

The work supports the stated conclusions. The methodology lacks certain details enlisted below.

I have the following suggestions to improve this manuscript:

Figure 1B: Why are the samples shown were collected on different day post-infection?

Figure 1E: The Y axis title should be CD4+ T cells in the duodenum, not colon.

Figure 2 and its legend: C-D and E-F are swapped. C-D should be control animals in the legend.

Figure 3 A: What parameters (method) were used to determine the monocyte count?

Supplementary Figure 3 and 4: the legend to the arc is present, but to the slices of pie chart is missing. Must be included in the figure.

Supplementary Figure 5 A-B the figure shows increase of Ki-67 expression in CD8 T cells in the duodenum, the text mentions the same increase in the colon.

Materials and Methods section:

What method was used to assess the function of T cells? In figure 4 they mention sorting, but there is no description in the method if it was intracellular cytokine staining or some other method.

Reviewer #3 (HIV immunity, NHP) (Remarks to the Author):

Le Hingrat and colleagues present results on the impact of depletion of CD4 T cells using a depleting CD4 rhesusized antibody (21 doses of CD4R1), in the SIVagm infected African Green Monkey model of HIV non-progression. The authors assessed the effects of this depletion on gut integrity, immune activation, immune cell populations, viral replication, including microbial translocation. The authors show that prolonged CD4R1 was not toxic and was effective in depleting CD4 T cells across the blood and to some degree the lymphoid tissue organs but had relatively little influence on gut integrity. The authors conclude the depletion was insufficient to explain the going hypothesis that this is driving gut disturbances and integrity and provide several alternate determinates in the discussion based on the findings in this model. These findings contribute much to the field in terms of understanding HIV-host (gut) interactions that remain complicated with few successful therapeutics.

Antiretroviral therapy is able to partially restore CD4 T cell counts on most complaint persons with viremic HIV infection with the notable exception of immune non responders. CD4 T cells loss and poor recovery are linked to numerous adverse pathological outcomes including residual inflammation, advancing disease progression and gut damage. The authors should be commended for methodically presented results and an expansive body of prior literature in the background to the study. Intriguing results demonstrate unexpectedly that a high magnitude of CD4 T cell depletion in blood and tissues was not sufficient to drive disease progression as monitored by extensive well thought out immunological, GI integrity and virological quantitative studies. The authors conduct robust detailed tissue studies in Lymph nodes, lower and upper GI tracts, and also at autopsy. Limitations is despite these findings no alternate pathways was investigated either experimental to reveal alternate mechanisms as the model is quite unique from the progressive disease setting with good control. Additional suggestions for improvement should be considered as outlined below:

Major concerns:

1/ Compensation by other cell types: Was the evaluation of bone marrow considered to delve deeper to assess the immunological changes that may further inform on the effective of CD4R1 to drive sufficient change to lead to gut related disturbances.

2/ A discussion on downregulated CD4 on the cell subsets was touched upon but should be expanded further on in the context of compensatory pathways in the unique AGM model and the effectiveness or lack of, the CD4R1 antibody on alternate undefined cell types. Will RNAseq comparative studies provide further clarification or insight?

3/ Studies on epithelial integrity such as transepithelial resistance would be useful to document if this was perturbed in the absence of CD4 cells.

4/ While granted this study is focused on using a non-progressive disease model, however the authors should explain in the discussions limitations of the model itself when taking into consideration the human HIV disease setting in the presence of ART.

Minor concerns:

Page 4 – CD4 depleting ab – Mention the clone used (ref 28)

Page 4 – virus uptake – ref 26 – include a statement in this model though other studies highlight alternate mechanisms

Page 4 - antibody-mediated depletion of CD4 T cell should change to CD4 expressing cells

Methods: Clone of CD4 T cell used for staining should be identified in the methods.

We would like to thank the Editorial Board, the Editor and the Reviewers for carefully reviewing our manuscript and their insightful and constructive comments that critically contributed to improve it.

Reviewer #1 (R1) (HIV, SIV, NHP models) (Remarks to the Author)

We would like to thank R1 for her/his appreciation of our work and the very insightful comments that contributed to improve our manuscript.

The authors have been working on the exact mechanism of AIDS progression in HIV/SIV infections. In particular, they have been working on a non-pathogenic model of SIV infection in African Green Monkeys (AGM) and a pathogenic model of SIV infection in Rhesus Monkeys (RM) (ref. 29-31). They have accumulated data indicating that gastrointestinal (GI) tract maintenance is the key for not developing AIDS in SIV-infected AGMs, while CD4 depletion could not be the main driver for AIDS progression. For this conclusion, further evidence should be accumulated. This study showing non-acceleration of AIDS progression by prolonged antibody-mediated CD4+ T cell depletion in SIV-infected AGMs adds an important data supporting the above conclusion, although unanswered questions still remain. Results on the prolonged antibody-mediated CD4+ T cell depletion are valuable. The following issues should be addressed.

1. One of the limitations of this study is reduction in viral loads (VL) by CD4 depletion, possibly by loss of SIV target cells. VL reduction could mitigate the impact of CD4 depletion itself on disease progression. Antibody-mediated CD4 depletion may be different from CD4 depletion under viral replication and CD4 restoration from acute CD4 depletion in GI. The authors should mention this point more clearly with more detailed discussion.

Thank you very much for this excellent suggestion. We agree that reduction in viral loads could potentially limit the impact of CD4⁺ T-cell depletion on disease progression and/or hinder disease progression. However, different data suggest that the level of viral replication is not the only factor contributing to disease progression. Several cases of disease progression have been reported in persons living with HIV (PWH) despite undetectable or low viremia (*Okulicz et al. JID, 2009; Sauce et al., Blood, 2012*). Also, in patients with very severe CD4⁺ T-cell depletion there is a trend to lower viral loads (*JAMA. 2006;296:1498-1506*). Conversely, high levels of viral replication are observed in natural hosts of SIV, which do not progress to AIDS; thus, mandrills and sooty mangabeys that do not usually progress to AIDS exhibit high viremia during chronic infection (between 10⁵ and 10⁶ copies/mL; *Pandrea et al., Virology, 2003; Silvestri et al., Immunity 2003*).

In our study, the CD4-depleting antibody treatment reduced CD4⁺ T cells below the levels observed in NHPs and PWH that progress to AIDS. Should the depletion of CD4-expressing cells be a major player in the progression to AIDS in natural hosts of SIV, one would expect that such a dramatic depletion of CD4⁺ T cells would result, at least, in an increase in the levels of markers associated with disease progression and/or opportunistic infections. Yet, the markers of disease progression (T-cell immune activation and inflammation) were not modified in our CD4-depleted NHPs. Interestingly, it has been recently reported that antibody-mediated CD4 depletion was not sufficient to reactivate latent tuberculosis in rhesus macaques, underlining that a CD4-independent mechanism could be responsible for the control of this latent infection (*Buçşan et al., JCI, 2019*). The absence of immune activation during the CD4-depleting antibody treatment is also important, as it has been reported that T-cell activation is a predictor of disease progression in PWH with low or undetectable viremia (*Hunt et al., Journal Inf Dis, 2008*), i.e., a population close to our CD4-depleted animals which have low plasma viral loads too.

At the R1 suggestion, we have extended the discussion in our revised manuscript (pages 20) to highlight this limitation of our study (reduction of viral loads):

“Throughout the follow-up, no significant gut dysfunction was observed in the CD4⁺ T-cell-depleted AGMs, none of which progressed towards AIDS, nor developed opportunistic infections. One limitation of our study is that the lower viral replication in CD4⁺ cell depleted NHPs could

have impacted disease progression. However, viral replication is not the sole factor contributing to disease progression, as progression to AIDS was reported to occur in individuals with low to undetectable viremia. In those patients, increases of the levels of soluble markers or immune activation preceded disease progression. No increase in markers of gut dysfunction (sCD14, IFABP, sCD163, LPS levels in tissues) or inflammation (CRP, sCD163) were seen in CD4⁺ T-cell-depleted AGMs during the follow-up. This suggests that prolonged CD4⁺ T-cell depletion is not fueling SIV-associated gut dysfunction, nor driving SIV disease progression. Furthermore, disease progression is usually occurring swiftly in NHPs, i.e., within a year. Hence, we believe it is unlikely that the AGMs in our study would have progressed to AIDS if we would have extended the follow-up beyond the 400 days. Together, these observations suggest that a profound and persistent CD4⁺ T-cell depletion can be tolerated without any major side effect and does not induce disease progression, if epithelial gut integrity is preserved, and chronic T-cell immune activation and systemic inflammation are kept at bay.”

As stated by the reviewer, cell death can occur through different mechanisms when it is triggered by a CD4-depleting antibody (apoptosis) or by viral replication (apoptosis, pyroptosis), the latter could be more likely to create an inflammatory environment. However, the magnitude of CD4⁺ T-cell depletion in AGMs treated with the CD4-depleting antibody was similar or higher than those reported in humans and NHPs progressing to AIDS. In our animals the virus continues to replicate and, as such, a fraction of CD4⁺ T cells could have also been depleted by the virus, thus limiting the difference between our experimental CD4⁺ T-cell depletion model and the models of progressive infection.

We have also specified this in the revised manuscript (*page 20*):

« Restoration of CD4-expressing cells might also be different in our model, in which there is no CD4⁺ T-cell restoration, there is no disease progression, and the progressive HIV/SIV infections, where progression occurs in spite of a modest mucosal CD4⁺ T cell restoration in the early stages of chronic infection.”

2. Related with the comment 1, VL reduction may result in reduction of immune activation. However, difference in immune activation between CD4-depleted and Controls was unclear. In Fig 4, CD8ab+IFN γ + (& CD8abGrzB+) was increased on d371 in CD4-depleted. Some discussion is required on these points.

We did not find statistical differences in the levels of immune activation (IA) between the two groups. We agree with R1 that lower VL could have indeed hindered such differences, but, in AGMs, IA is usually disconnected from the level of viral replication during the chronic infection. This feature is also found in other natural hosts of SIV (e.g., sooty mangabeys and mandrills) (Silvestri et al., *Immunity*, 2003; Onanga et al., *J Virol*, 2006)

Moreover, if the lower VLs were the cause of the lower IA, a reduced IA should be expected in CD4-depleted AGMs on samples collected between D28 and D42 when the pVL were >1 log lower than in controls (i.e., after initiating the CD4-depleting antibody treatment and before reaching the set-point viral replication during natural infection), which was not the case.

R1 is correct that expression of IFN- γ or granzyme B were slightly higher at D371 in the CD4-depleted AGMs than in controls, but this difference was not significant. There was no significant increase in the percentage of CD8 $\alpha\beta$ T cells expressing IFN- γ or granzyme B overtime. We have added a sentence in the Results section to specify this (*page 10*):

“No significant differences were observed between controls and CD4-depleted AGMs with regard to the fraction of cells expressing specific functional markers.”

3. It remains unclear whether prolonged antibody-mediated CD4+ T cell depletion accelerates AIDS progression in SIV-infected RMs.

We agree with R1 that it is not yet known whether the antibody-mediated CD4⁺ T-cell depletion accelerates progression to AIDS in SIV-infected RMs. In the existing literature, the NHPs that

progressed faster to AIDS after antibody-mediated CD4 depletion had received the CD4-depleting antibody treatment before SIV infection, which could have affected both the immune responses and the viral tropism during the virus seeding phase.

In addition, previous reports showed difference in type I IFN responses in SIV infection between AGMs and RMs. It is recommended to have a discussion on these points.

This is an excellent suggestion. Indeed, we had not previously discussed this other important particularity of natural hosts of SIV. It is possible that once animals have been able to mount an efficient type I IFN response (and an efficient immune response in general), the depletion of CD4⁺ T cells could have been insufficient to revert their ability to control SIV infection.

This cannot be evaluated in our study, as the CD4 depletion was performed at 21 DPI, to avoid interferences with the seeding of the viral reservoir and with the depletion of CD4⁺ T cells naturally occurring during the acute infection.

We discussed these issues in the revised version (*page 20*):

“However, it cannot be excluded that other mechanisms might be involved in the protection of AGMs against disease progression, including the ability to mount a more efficient acute type I IFN response, and that the depletion of CD4⁺ T cells might have been insufficient to revert their ability to control SIV infection. The role of this efficient immune response of natural hosts of SIV could not be assessed in this study, as we initiated the CD4-depleting antibody treatment at 21 dpi, i.e., after the innate immune response, to avoid altering virus reservoir seeding during acute infection.”

4. In Fig 2B, individual means of Controls is expected to be 100, but data shown are not. Please explain the reason.

In Figure 2A-B of the original submission, the bars represented the median, while the mean was indeed at 100%, as expected. We have updated the Figure 2, with the bars now representing the means in panels A and B.

5. CD4 depletion may result in change in viral tropism. It would be better to show data on env sequences and/or viral tropisms.

The R1 has an excellent point. We agree that CD4 depletion could have induced a shift in viral tropism. This has notably been described in animals in which CD4 depletion was initiated before SIV infection (Ortiz et al., JCI, 2012; Micci et al., Plos Path, 2014). To investigate this hypothesis, we have sequenced the full *env* coding sequences from plasma samples collected 21 dpi (i.e., right before initiating CD4 depletion) and at the necropsy, in both control and CD4-depleted AGMs. We aligned those 21 *env* sequences (one per animal at D21, plus 5 from CD4-depleted animals and 4 from controls at time of necropsy, all of which have been deposited on Genbank, under the accession numbers OP491432-OP491452) with 15 other sequences (including the reference strain SIVsab92018) that were available in the GenBank.

When analyzing the sequences obtained, along with the 15 others full *env* sequences, the *env* sequences of the strains collected post-CD4 depletion did not differ from those amplified from plasma collected before initiating CD4 depletion (i.e., 21 days postinfection), and that the sequences from the AGMs in the two groups clustered together and were interspersed in the phylogenetic tree built with a maximum likelihood analysis. Should the CD4 depletion have induced a shift in viral tropism, a divergent evolution of *env* sequences would have been observed, with *env* sequences from postdepletion samples clustering apart from predepletion ones. Instead, we observed that, both at 21 dpi and at the time of necropsy, *env* sequences were similar between CD4-depleted and control AGMs, and that all sequences were genetically close to the parental strain SIVsab92018 (mean genetic distance to the reference strain, SIVsab92018: 0.0045 substitutions per base) (Figure S7-B). As predepletion and postdepletion *env* sequences are close

to each other in the phylogenetic tree (Figure S7A), this suggests that CD4 depletion did not induce a bias in viral evolution.

We next inspected the *env* sequences to see if asparagine-linked glycosylation sites (NGS) were altered. NGS are important for CD4 use, as highlighted by the cases of acquired resistance to ibalizumab (a monoclonal antibody targeting the domain 2 of the CD4, used to treat multidrug resistant HIV-1 infection) that have been linked to the loss of a N-glycosylation site in the V5 loop (Toma et al., *J Virol*, 2011; Pace et al., *JAIDS*, 2013). The importance of those NGS has also been shown in SIVmac (Yen et al., *J Virol*, 2014), for which the loss of the N173 in the V2 loop led to an enhanced macrophage tropism (Yen et al., *J Virol*, 2014). No mutation, nor deletion of the potential NGS were seen in the *env* sequences, either before or after CD4 depletion (Figure S7C)

We also looked at CD4 binding sites throughout the *env* sequences, and we did not observe any change in amino acids on those binding sites. Furthermore, neither insertions, nor deletions were observed in the V3 loop of those viruses (Figure S7C), and the net charge of the V3 loop was unchanged in all sequences (total charge: 6). Insertions and a gain in positively charged amino acids are two mechanisms that have been linked to a switch from a R5 to a X4 tropism in HIV-1 and HIV-2 (De Jong et al., *J. Virol.*, 1992; Visseaux et al., *J. Inf. Dis.*, 2012).

Another argument for the absence of change in the viral tropism is that viral reservoir and VLs did not increase in the central nervous system (Figure 10), while this was observed by Micci et al. in RMs whose viruses had evolved to have CD4-independent envelopes.

These sequence analyses have been added to the revised version and are now discussed in the Results section (pages 15-16):

“The CD4⁺ T-cell depletion did not result in a divergent evolution of the *env* sequences amplified from the plasma of the CD4⁺-depleted AGMs, compared to the ones from controls (Supplementary Fig. S7A). The mean genetic distance between the parental strain, SIVsab92018, and the *env* sequences were similar between the 2 groups, at 21 dpi (i.e., before initiating the CD4-depleting antibody treatment) and at time of necropsy (Supplementary Fig. S7B). Furthermore, when visually inspecting the *env* sequences, CD4-depleted animals did not exhibit any change in the functional sites associated with the emergence of CD4-independent envelopes (loss of asparagine-linked glycosylation sites) or with a switch from a R5 to a X4 tropism (insertions and/or change in the net charge of the V3 loop) (Supplementary Fig. S7C).”

6. Ref 28 should be corrected (lacking in page number?).

Thank you for pointing out this, we have now updated the Ref 28 (Swanstrom et al., *JCI*, 2021).

R2 (HIV therapy, SIV models) (Remarks to the Author)

We would like to thank R2 for her/his appreciation to our work and the very insightful comments that contributed to improve our manuscript.

Breakdown of the gut mucosal integrity is one of the hallmarks of HIV infection. Several factors may be involved in this intestinal damage. Central amongst them is the profound loss of mucosal CD4⁺ T cells, that have key role in the control of systemic breakthrough by enteral bacteria. African green monkeys are natural hosts of SIVsab. They do not show signs of intestinal damage even in the presence of robust SIV proliferation. Le Hingrat et al. investigated if prolonged CD4⁺ T cell depletion would elicit breakdown of the gut mucosal integrity in a cohort of SIVsab-infected African Green Monkeys. Although the depletion was nearly 90% in the intestine, (and even higher in other tissues and the blood) they were not able to find any sign of deterioration of intestinal integrity, chronic activation of the immune system, or functional takeover of the role of CD4 T cells

by remaining T cell populations. The significance of this finding is that we may not need the total restoration of the mucosal CD4 T cell pool to maintain the integrity of the intestinal barrier. Perhaps there are other factors that could be targeted by novel therapeutics. However, the questions still remain:

- is 90% depletion leaves enough CD4 T pool in critical mucosal compartments that one year follow-up is not enough to achieve the detrimental effect,
- can the mechanism that renders the African green monkeys refractory against the disease progression be found in humans as well?

Indeed, the two main questions raised by R2 are some of the limitations of this work. It is possible that the effects of CD4 depletion could only appear after an extended period of time or that the rare CD4⁺ T cells left are sufficient to protect from disease progression. However, in humans, CD4 depletion is not complete in tissues either. Moreover, RMs with similar, or less CD4 depletion in circulation and at the mucosal sites than our CD4-depleted AGMs usually progress to AIDS within a year. We have followed the AGMs with severe CD4 depletion for >400 days, and we have not observed any increase in markers that are associated with disease progression, making it unlikely that they would have progressed to AIDS within the following few months.

This has been also specified in the revised manuscript (*pages 18 and 20*):

“One strength of our study is the magnitude of the CD4⁺ T-cell depletion achieved in tissues, at least equivalent to what is observed in humans and NHPs that progress to AIDS.”

“Furthermore, disease progression is usually occurring swiftly in NHPs, i.e., within a year. Hence, we believe it is unlikely that those AGMS would have progressed to AIDS after >400 days follow-up.”

Regarding the mechanism that render AGMs refractory to disease progression, different mechanisms have been hypothesized, that might be intertwined: a more efficient immune response during early infection, maintenance of gut integrity during chronic infection, and the presence of other immune cells that compensate the loss of CD4⁺ T cells.

This is now more extensively discussed in the revised version (*page 20*):

“Throughout the follow-up, no significant gut dysfunction was observed in the CD4⁺ T-cell-depleted AGMs, none of which progressed towards AIDS, nor developed opportunistic infections. One limitation of our study is that the lower viral replication in CD4⁺ cell depleted NHPs could have impacted disease progression. However, viral replication is not the sole factor contributing to disease progression, as progression to AIDS was reported to occur in individuals with low to undetectable viremia.⁵³ In those patients, increases of the levels of soluble markers or immune activation preceded disease progression. No increase in markers of gut dysfunction (sCD14, IFABP, sCD163, LPS levels in tissues) or inflammation (CRP, sCD163) were seen in CD4⁺ T-cell-depleted AGMs during the follow-up. This suggests that prolonged CD4⁺ T-cell depletion is not fueling SIV-associated gut dysfunction, nor driving SIV disease progression. Furthermore, disease progression is usually occurring swiftly in NHPs, i.e., within a year. Hence, we believe it is unlikely that the AGMS in our study would have progressed to AIDS if we would have extended the follow-up beyond the 400 days. Together, these observations suggest that a profound and persistent CD4⁺ T-cell depletion can be tolerated without any major side effect and does not induce disease progression, if epithelial gut integrity is preserved, and chronic T-cell immune activation and systemic inflammation are kept at bay. However, it cannot be excluded that other mechanisms might be involved in the protection of AGMs against disease progression, including the ability to mount a more efficient acute type I IFN response, and that the depletion of CD4⁺ T cells might have been insufficient to revert their ability to control SIV infection. The role of this efficient immune response of natural hosts of SIV could not be assessed in this study, as we initiated the CD4-depleting antibody treatment at 21 dpi, i.e., after the innate immune response, to avoid altering virus reservoir seeding during acute infection.”

As shown by a preliminary study (*Hao et al., Nat Comm, 2015*), the loss of gut integrity through administration of DSS (to induce colitis) caused an increase in soluble markers of inflammation

and in IA in 2 AGMs. Even if other mechanisms could be involved in controlling disease progression, they are not sufficient to counter the negative impact of the gut damage on HIV/SIV pathogenesis. This hints that preserving gut integrity could be beneficial to humans too, in which damage to the gut is reported to persist even in patients controlling viral replication with ART.

This has also been added to the revised version (*pages 21-22*):

“While different mechanisms could be involved in the protection of AGMs against AIDS, in a preliminary study, we showed that damages to the gut integrity in SIV-infected AGMs resulted in an increase in markers associated with disease progression. This suggests that gut damage can override the effects of other mechanisms preventing disease progression in the natural hosts of SIVs, and that preservation of gut integrity could be beneficial to humans too.”

The work supports the stated conclusions. The methodology lacks certain details enlisted below. I have the following suggestions to improve this manuscript:

Figure 1B: Why are the samples shown were collected on different day post-infection?

We showed flow data from samples collected at different dates postinfection, as the two types of intestinal biopsies were not collected at the same timepoint. We were alternating colonic and duodenal biopsies during this study. For consistency, in the updated **Figure 1**, we are now showing the data for 42 dpi for both whole blood and colonic biopsies in panel B to have matching timepoints and because 42 dpi corresponds to 21 days post-CD4 depletion, and it is the first timepoint on which the CD4 depletion is severe in the gut. Data for 42 and 48 dpi can still be seen in the panel C of **Figure 1**. D84 was chosen for duodenal and lymph node biopsies, as it is the first postinfection sample that is available for both.

Figure 1E: The Y axis title should be CD4⁺ T cells in the duodenum, not colon.

Figure 2 and its legend: C-D and E-F are swapped. C-D should be control animals in the legend.

We apologize for these mistakes. Thank you for pointing them out. We have updated the corresponding figures in the revised version.

Figure 3 A: What parameters (method) were used to determine the monocyte count?

We determined the exact number of monocytes using TruCount tubes that contained a mix of antibodies (including CD14 and CD16). We gated first by size and by CD45 expression, then with CD14 and CD16. Monocytes were defined as being CD14⁺ or CD16⁺.

This has now been specified in the revised Material and Methods section (*pages 25-26*).

“First, TruCount staining was performed on 50 μ L of whole blood, using CD45 (PerCP-Cy5-5, clone D058-1283), CD3 (V450, SP34-2), CD14 (PE-Cy7, M5E2), CD16 (APC-Cy7, 3G8) and CD163 (APC, GHI/61) antibodies. This allowed to precisely quantify CD45⁺ cells, T cells, and monocytes in blood. The exact counts of CD4⁺ and CD8⁺ T-cells were determined by multiplying the exact number of CD3⁺ T cells by the percentage of CD4⁺ or CD8⁺ cells among CD3⁺ T cells that were determined in a different staining. CD14 and CD16 were used to identify circulating monocyte subsets.”

An example of the gating strategy is also shown in **Supplementary Figure S8**.

Supplementary Figure 3 and 4: the legend to the arc is present, but to the slices of pie chart is missing. Must be included in the figure.

We have updated these legends. The same color code was used for both the arcs and the pie slices.

Supplementary Figure 5 A-B the figure shows increase of Ki-67 expression in CD8 T cells in the duodenum, the text mentions the same increase in the colon.

This has been clarified in the updated version (page 12):

“No differences were observed between groups regarding CD8⁺ T-cell proliferation in colon biopsies, while in duodenal biopsies there was an increase in Ki-67 expression by CD8⁺ T cells at 21 and 42 dpi in controls, which was not observed in CD4⁺ T-cell-depleted AGMs”.

Materials and Methods section:

What method was used to assess the function of T cells? In figure 4 they mention sorting, but there is no description in the method if it was intracellular cytokine staining or some other method.

We have added the details of the intracellular cytokine staining method used to the Material and Methods section. The cells were not sorted prior to ICS staining, this typo has been removed from the title and legend of the **Figure 4**.

The following paragraph has been added to the Methods section (pages 26-27):

“Intracellular Cytokine Staining

Frozen cells were thawed, washed, and split into two tubes containing each approximately 2x10⁶ cells in 1 mL of complete RPMI media. Cells in one tube were stimulated with 5 ng/ml PMA and 1 µg/ml ionomycin. Both tubes were treated with 10 ng/ml of brefeldin A, 1µl/ml monensin (BD GolgiStop), and CD107a (BV711, clone H4A3) for 16-18 hours at 37°C. Cells were washed twice in PBS and stained with fixable Live/Dead Aqua, CD3 (ALX700, SP34-2), CD4 (BV605, L200), CD8 (PerCP-Cy5.5, RPA-T8), CD28 (ECD, CD28.2) and CD95 (Cy5PE, DX2) for 20 min at 4°C. Cells were washed with PBS and permeabilized with Foxp3 Perm Solution (eBioscience) for 1 hr at 4°C. Cells were washed twice with 2 ml FoxP3 Perm Wash Buffer Solution (eBioscience) and stained with FoxP3 (Alx488, 206D), GranzymeB (PE, GB11), IFNα (Cy7PE, 4S.B3), IL-2 (BV785, MQ1-17H12), IL-17 (e450, eBio64DEC17), and CD40L (APC-e780, 24-31) for 30 min at 4°C. Cells were washed twice with 2 ml FoxP3 Perm Wash Buffer Solution and fixed in 1% PFA before being analyzed on a LSRFortessa. Unstimulated cells were used to set gates for stimulated cells from the same animal and timepoint to account for background staining. Data was analyzed using FlowJo® v10.7.1, Graphpad Prism® v8.4.3 (GraphPad), and SPICE 6. Data was only included for analysis if at least 200 cells were present in the target population.”

R3 (HIV immunity, NHP) (Remarks to the Author)

We would like to thank R3 for her/his appreciation to our work and the very insightful comments that contributed to improve our manuscript.

Le Hingrat and colleagues present results on the impact of depletion of CD4 T cells using a depleting CD4 rhesusized antibody (21 doses of CD4R1), in the SIVagm infected African Green Monkey model of HIV non-progression. The authors assessed the effects of this depletion on gut integrity, immune activation, immune cell populations, viral replication, including microbial translocation. The authors show that prolonged CD4R1 was not toxic and was effective in depleting CD4 T cells across the blood and to some degree the lymphoid tissue organs but had relatively little influence on gut integrity. The authors conclude the depletion was insufficient to explain the going hypothesis that this is driving gut disturbances and integrity and provide several alternate determinates in the discussion based on the findings in this model. These finding contribute much to the field in terms of understanding HIV-host (gut) interactions that remain complicated with few successful therapeutics.

Antiretroviral therapy is able to partially restore CD4 T cell counts on most complaint persons with viremic HIV infection with the notable exception of immune non responders. CD4 T cells loss and poor recovery are linked to numerous adverse pathological outcomes including residual inflammation, advancing disease progression and gut damage. The authors should be commended for methodically presented results and an expansive body of prior literature in the background to the study. Intriguing results demonstrate unexpectedly that a high magnitude of CD4 T cell depletion in blood and tissues was not sufficient to drive disease progression as monitored by extensive well thought out immunological, GI integrity and virological quantitative studies. The authors conduct robust detailed tissue studies in Lymph nodes, lower and upper GI tracts, and also at autopsy. Limitations is despite these findings no alternate pathways was investigated either experimental to reveal alternate mechanisms as the model is quite unique from the progressive disease setting with good control. Additional suggestions for improvement should be considered as outlined below:

Major concerns:

1/ Compensation by other cell types: Was the evaluation of bone marrow considered to delve deeper to assess the immunological changes that may further inform on the effective of CD4R1 to drive sufficient change to lead to gut related disturbances.

Unfortunately, we did not collect cells from bone marrow during necropsies. In a previous study (Micci et al, Plos Pathogens, 2014), it was shown that CD4⁺ T cells were also depleted in the bone marrow of CD4R1-receiving RMs. It was also reported that disease progression is linked to exhaustion of lymphopoiesis (Douek et al., Nature, 1998; Sauce et al., Blood, 2011). We agree with the reviewer that it would have been extremely interesting to study in depth the impact of the CD4-depleting treatment on the different immune progenitors in the bone marrow, to quantify the impact of this treatment on lymphopoiesis and to determine if it is similar to what is seen during a progressive infection. This has now been specified in the revised version of the manuscript (page 21):

“Restoration of CD4-expressing cells might also differ between our model combining viral replication and CD4⁺ T-cell depletion and a model of progressive infection in which individuals or animals receive ART. The impact on lymphopoiesis might differ, and it has been previously shown that exhaustion of lymphopoiesis can drive disease progression.”

2/ A discussion on downregulated CD4 on the cell subsets was touched upon but should be expanded further on in the context of compensatory pathways in the unique AGM model and the effectiveness or lack of, the CD4R1 antibody on alternate undefined cell types. Will RNAseq comparative studies provide further clarification or insight?

We agree with the R3 that alternate undefined cell types could also play a role in the protection of AGMs against disease progression. RNAseq could be performed in a future study, but has to be done on whole blood, to be able to explore all immune cell types, and not only **PBMC, which are the only cells that we are routinely isolating in our laboratory.**

3/ Studies on epithelial integrity such as transepithelial resistance would be useful to document if this was perturbed in the absence of CD4 cells.

Unfortunately, those experiments require either freshly collected intestinal samples or culturing intestinal organoids from intestinal crypt stem cells, which is not done routinely in our laboratory. All biopsies that we have been collected were directly used to separate lymphocytes, and thus we cannot perform this experiment, which could indeed have brought interesting insights on the epithelial integrity in AGMs that have received or not the CD4-depleting antibody. However, the surrogate markers used in our study (plasma I-FABP and claudin expression in colon) indicate that gut permeability was not affected by the CD4 depletion.

4/ While granted this study is focused on using a non-progressive disease model, however the authors should explain in the discussions limitations of the model itself when taking into consideration the human HIV disease setting in the presence of ART.

We agree with the R3 that many potential mechanisms of protection of natural hosts of SIV from disease progression have been discussed in the literature, and that not all of them could be leveraged in PWH. For example, immune responses and gut integrity could be improved in PWH receiving ART with new therapeutics, but if the presence of specific immune cell subsets is critical, this might not be possible to alter in PWH if those immune cells are absent or negligible in humans. Fortunately, the role of those other immune cells seems to be limited in AGMs, as they are neither expanding, nor gaining functionality during the CD4 depletion treatment. Moreover, preliminary data from a study in which DSS was administered to AGMs (*Hao et al., Nature Communications, 2015*) showed that the damage to the gut integrity overrides any other potential mechanisms of protection, as shown by the increase in multiple markers associated with disease progression. This suggests that maintenance of gut integrity could be the most important protection factor, and it is also reassuring because restoring gut integrity could be achievable in patients treated with ART.

This has been specified in the revised manuscript (*page 22*):

"While different mechanisms could be involved in the protection of AGMs against AIDS, in a preliminary study, we showed that damages to the gut integrity in SIV-infected AGMs resulted in an increase in markers associated with disease progression. This suggests that gut damage can override the effects of other mechanisms preventing disease progression in the natural hosts of SIVs, and that preservation of gut integrity could be beneficial to humans too."

We have extended the discussion in the revised version of the manuscript to detail the other mechanisms of protection that could exist in natural hosts of SIV (*page 20*):

"However, it cannot be excluded that other mechanisms might be involved in the protection of AGMs against disease progression, including the ability to mount a more efficient type I IFN response, and that the depletion of CD4⁺ T cells might not revert their ability to control SIV infection. The role of this efficient immune response of natural hosts of SIV could not be assessed in this study, as we initiated the CD4-depleting antibody treatment at 21 days postinfection, i.e., after the innate immune response, to avoid altering virus reservoir seeding during primary infection."

We have also included a new paragraph on the limitation of this model when comparing it to the PWH

"Restoration of CD4-expressing cells might also differ between our model combining viral replication and CD4⁺ T-cell depletion and a model of progressive infection in which individuals or animals receive ART. The impact on lymphopoiesis might differ, and it has been previously shown that exhaustion of lymphopoiesis can drive disease progression. Furthermore, our model might not recapitulate other characteristics of pathogenic HIV/SIV infections, in which multiple immune cell types are dysfunctional due to the persistent immune activation and inflammation that persist even during ART."

Minor concerns:

Page 4 – CD4 depleting ab – Mention the clone used (ref 28)

Thank you for pointing out this. This is now specified in the text (*Page 6*). The clone used is OKT4A.

Page 4 – virus uptake – ref 26 – include a statement in this model though other studies highlight alternate mechanisms

In animals that were experimentally depleted in CD4⁺ T cells before being infected, it has been reported that mainly non-T cells (microglia, macrophages) were infected (*Micci et al., Plos Path, 2011*). Although Silvestri and colleagues showed the emergence of CD4-independent envelopes in sooty mangabeys (*Ortiz et al., JCI, 2011*), we did not see a change in viral tropism in our AGMs. This was highlighted by the limited genetic evolution of *env* sequences after over a year of antibody-mediated CD4 depletion. Furthermore, we did not see an increase in the levels of cell-associated viral RNA and DNA in brain tissues, contrary to previous reports (**Figure 10**). All this indicates that, in our model, contrary to the RMs that received the CD4 depleting antibody treatment before being infected with SIV, the virus uptake was not modified.

This is likely due to the fact that we have initiated CD4 depletion once the reservoirs were established, and thus we did not have a bias towards the selection of CD4-independent envelopes during the primary infection.

We modified the text accordingly (*page 4*):

“Experimental CD4⁺ T-cell depletion in rhesus macaques (RMs) prior to SIV infection abrogated the postacute control of viremia, led to the emergence of CD4-independent envelopes and to a larger number of non-T cells infected cells, and accelerated disease progression.”

Page 4 - antibody-mediated depletion of CD4 T cell should change to CD4 expressing cells

Thanks for pointing this to us. This has been modified in the revised version.

Methods: Clone of CD4 T cell used for staining should be identified in the methods.

The clone used for CD4 staining is L200, this has been specified in the methods of the revised manuscript (*Page 26*).

REVIEWERS' COMMENTS

Reviewer #1 (Remarks to the Author):

The authors adequately responded to all of my comments.

Reviewer #2 (Remarks to the Author):

Breakdown of the gut mucosal integrity is one of the hallmarks of HIV infection. Several factors may be involved in this intestinal damage. Central amongst them is the profound loss of mucosal CD4+ T cells, that have key role in the control of systemic breakthrough by enteral bacteria. African green monkeys are natural hosts of SIVsab. They do not show signs of intestinal damage even in the presence of robust SIV proliferation. Le Hingrat et al. investigated if prolonged CD4+ T cell depletion would elicit breakdown of the gut mucosal integrity in a cohort of SIVsab-infected African Green Monkeys. Although the depletion was nearly 90% in the intestine, (and even higher in other tissues and the blood) they were not able to find any sign of deterioration of intestinal integrity, chronic activation of the immune system, or functional takeover of the role of CD4 T cells by remaining T cell populations. The significance of this finding is that we may not need the total restoration of the mucosal CD4 T cell pool to maintain the integrity of the intestinal barrier. Perhaps there are other factors that could be targeted by novel therapeutics. However, the questions still remain: is 90% depletion leaves enough CD4 T pool in critical mucosal compartments that one year follow-up is not enough to achieve the detrimental effect, can the mechanism that renders the African green monkeys refractory against the disease progression be found in humans as well?

The manuscript in its current form supports the stated conclusions. The methodology is sound, and the work can be reproduced based on the details provided in the Method section.

Reviewer #3 (Remarks to the Author):

The authors have satisfactorily addressed several concerns, Namely: Given the limitations of additional tissue specimen access the responses provided are acceptable. Glad to read now clarification in the discussion on alternate mechanisms(though speculative) for consideration that may compensate for CD4 T cell loss. Overall the responses are acceptable by this reviewer.